# Insight into the bioactivity and action mode of betulin, a candidate aphicide from plant metabolite, against aphids

**Junxiu Wang[1†], Matthana Klakong[2†], Qiuyu Zhu[2], Jinting Pan[1], Yudie Duan[1], Lirong Wang[3], Yong Li[3], Jiangbo Dang[1], Danlong Jing[1], Hong Zhou[2,4*]**

[1]Key Laboratory of Agricultural Biosafety and Green Production of Upper Yangtze River (Ministry of Education), Chongqing Key Laboratory of Forest Ecological Restoration and Utilization in the Three Gorges Reservoir Area, College of Horticulture and Landscape Architecture, Southwest University, Chongqing, China; [2]College of Plant Protection, Southwest University, Chongqing, China; [3]Zhengzhou Fruit Research Institute, Chinese Academy of Agricultural Sciences, Zhengzhou, China; [4]Yibin Academy of Southwest University, Yibin, China

**\*For correspondence:**
zhouhong1990@swu.edu.cn

†These authors contributed equally to this work

**Competing interest:** The authors declare that no competing interests exist.

## eLife Assessment

This **important** study identifies a plant-derived metabolite, betulin, as an effective natural insecticide against aphids and uncovers its specific molecular target. The evidence is **compelling**, combining greenhouse and field efficacy trials with rigorous molecular, genetic, and electrophysiological approaches that converge on a conserved binding site in the aphid GABA receptor. While additional work is needed to fully assess potential off-target effects and ecological safety, the study provides a strong mechanistic foundation. These findings will be of interest to researchers in plant biology, chemical ecology, and sustainable pest management.

**Abstract** Pest-resistant plants usually utilize secondary metabolites to cope with insect infestation. Betulin, a key bioactive compound in aphid-resistant wild peach, possesses promising applications in crop protection. Here, betulin, in both greenhouse and field experiments, displayed excellent control efficacy against *Myzus persicae*. RNA-seq, quantitative real-time PCR (qRT-PCR), and western blotting revealed that betulin significantly inhibited the expression of *MpGABR* (encoding a GABA$_A$ receptor). Besides, RNAi-mediated silencing of *MpGABR* markedly increased aphid sensitivity to betulin. Furthermore, microscale thermophoresis (MST) and voltage-clamp assays indicated that betulin bound to MpGABR ($K_d$ = 2.24 μM) and acted as an inhibitor of MpGABR. Molecular docking, mutagenesis, and genome editing suggested that THR228 is a critical and highly conserved site in MpGABR that betulin binds to specifically, causing aphid death. Overall, the activity of betulin depends on specific targeting and inhibition of MpGABR. Elucidating the mechanism of action of this peach-derived insecticide may offer a sustainable green strategy for aphid control.

## Introduction

Over more than 400 million years of coevolution with herbivores, plants have developed a variety of morphological (*Soujanya et al., 2023*), biochemical (*Zhou et al., 2024b*), and molecular (*Wang et al., 2024*) defense strategies against insect herbivore attack. Plant secondary metabolites are

**eLife digest** The peach aphid is one of the most harmful pests worldwide and can parasitize over 400 species of plants. In addition to sucking phloem sap and secreting honeydew, the peach aphid can also spread various plant viruses, causing severe crop yield reductions and huge economic losses.

So far, the control of peach aphids mainly relies on chemical pesticides. However, the long-term and unreasonable use of these chemicals has led to environmental pollution and prompted aphids to develop resistance to pesticides, making the control of these insects even more difficult. Therefore, it is imperative to develop new, environmentally friendly insecticides.

For example, secondary plant metabolites, such as betulin – which is a key bioactive compound in aphid-resistant wild peach – have been proven to be highly toxic to aphids, demonstrating significant potential for development as a green insecticide. To find out how exactly betulin acts against aphids, Wang et al. used a combination of field trials and molecular, genetic and electrophysiological approaches.

The results showed that betulin inhibits the expression of *MpGABR* – a gene that helps regulate nerve signals in aphids – while simultaneously suppressing the function of the MpGABR protein by binding to its specific structural component called THR228. Exposing aphids to betulin significantly inhibited the expression of *MpGABR*. Micro-scale thermophoresis (a technique used to measure the binding between molecules) and voltage clamp experiments (used to study how compounds affect ion channel activity) confirmed that betulin could bind to the MpGABR protein via THR228 and acted as its inhibitor. Molecular docking (a computer method that predicts how a molecule interacts with a protein), mutagenesis analysis, and genome editing studies further suggested that THR228 is a key site that has remained nearly unchanged in aphid through evolution because it is essential for the protein's function.

The findings of Wang et al. will be beneficial for pesticide companies for developing new products, helping farmers control aphids and avoiding environmental pollution. More research is needed before these benefits can be realized, such as solvent formulation trials for developing betulin as an aphid insecticide, regional trials for using betulin as an aphid insecticide, and screening for insecticides with similar functions to this metabolite.

highly valuable natural compounds, offering abundant and potent biochemical defense driven by plant-insect coevolution (*Jahan et al., 2025*). Interestingly, wild relatives of crops with insect resistance usually accumulate large amounts of biologically active compounds (*Alonso-Salces et al., 2022*; *Wang et al., 2022a*). Accordingly, investigating the bioactivities and potential mechanisms of action of compounds derived from wild resistant germplasms may provide effective and sustainable strategies for pest control in crop protection.

*Myzus persicae* (Sülzer) is one of the most destructive sap-feeding pests worldwide. It is able to settle on more than 400 plant species globally, displaying extraordinary polyphagia and a wide range of hazards (*Bass et al., 2014*). In addition to ingesting phloem saps and secreting honeydew, *M. persicae* can transmit various plant viruses (*Guo et al., 2022*), resulting in severe losses in crop yields. To date, the principal control strategy for *M. persicae* has relied on synthetic insecticides, including pyrethroids, organophosphorus compounds, carbamates, and neonicotinoids (*Jiménez-Jiménez et al., 2019*; *Martins et al., 2021*; *Zhang et al., 2023*). However, long-term irrational application of these chemical insecticides has led to environmental pollution and promoted the resistance of aphids to insecticides (*Troczka et al., 2021*; *Zhang et al., 2023*; *Stará et al., 2024*). Therefore, it is necessary to develop novel green insecticides as potential alternatives to synthetic insecticides. Fortunately, plant secondary metabolites are a 'treasure trove of active compounds' that serve as a rich resource for the development of promising green insecticides.

Our previous study revealed that *Prunus davidiana*, a close wild relative of cultivated peach, displays strong resistance to *M. persicae* through the accumulation of high levels of betulin (*Wang et al., 2022a*; *Wang et al., 2024*). In addition, betulin, a lupane-type triterpene, possesses strong insecticidal activity and is a promising substance for the development of novel aphid-control insecticides. A number of studies have reported that betulin and its derivatives exhibit a wide range of pharmacological activities (*Amiri et al., 2020*). An immune stimulant, Ir-Bet, was prepared using iridium complex

and betulin, which evoked ferritinophagy-enhanced ferroptosis, thereby activating anti-tumor immunity (*Lv et al., 2023*). The anti-inflammatory effect of betulin has been reported in macrophages at lymphoma sites in mice (*Szlasa et al., 2023*). Betulin has been found to improve hyperlipidemia and insulin resistance and decrease atherosclerotic plaques by inhibiting the maturation of sterol regulatory element-binding protein (*Tang et al., 2011*). Additionally, betulin and its derivatives have been found to exhibit insecticidal activity against *Plutella xylostella* L. (*Huang et al., 2025*), *Aedes aegypti* (*de Almeida Teles et al., 2024*), and *Drosophila melanogaster* (*Lee and Min, 2024*). However, the insecticidal mechanism of betulin against aphids remains unclear.

Gamma-aminobutyric acid (GABA) receptors have been confirmed to be targets of terpenoids that impair neuronal function in insect herbivores (*Guo et al., 2023*). There are two types of GABA receptors: ionotropic (GABA$_A$) and metabotropic (GABA$_B$) receptors. The first GABA receptor subunit identified in insects is encoded by *Rdl*, which confers resistance to dieldrin in *D. melanogaster* (*Ffrench-Constant et al., 1991*). Although a variety of terpenoids play roles as positive allosteric modulators or noncompetitive antagonists (NCAs) of GABA$_A$ receptors (GABRs) (*Guo et al., 2023*), whether betulin targets the GABR in *M. persicae* needs to be further investigated. GABRs are heteropentameric ligand-gated ion channels in the central nervous system that conduct chloride and bicarbonate ions and are the target of numerous drugs for the treatment of neuropsychiatric disorders (*Thompson, 2024*). GABRs are composed of five different types of subunits, each with four helical transmembrane domains (TM1-TM4), of which TM2 is located at the center of the pentamer and forms an ion channel (*Fan et al., 2024*). A previous study has proposed that vertebrate and human GABR genes maintain a broad and conservative gene clustering pattern, while in invertebrates, this pattern is missing, indicating that these gene clusters were formed early in vertebrate evolution and were established after invertebrates diverged. Notably, invertebrates each possess a unique GABR gene pair, which is homologous with human GABR α and β subunits, suggesting that the existing GABR gene cluster evolved from an ancestral α-β subunit gene pair (*Tsang et al., 2007*). During the coevolution of plants and insects, the duplications and amino acid substitutions in GABR may have been beneficial for adaptation to insecticides and terpenoid compounds (*Guo et al., 2023*).

In this study, betulin was confirmed to have excellent control efficacy against *M. persicae* in both greenhouse and field experiments. RNA-seq, quantitative real-time PCR (qRT-PCR), and western blotting assays revealed that betulin significantly inhibited the expression of *MpGABR* in aphids. In addition, RNAi-mediated silencing of *MpGABR* markedly increased aphid sensitivity to betulin. Furthermore, microscale thermophoresis (MST) and voltage-clamp assays indicated that betulin can bind to MpGABR ($K_d$ = 2.24 μM) and act as an inhibitor (EC$_{50}$=20.66 μM) of MpGABR. Molecular docking analysis suggested that betulin bound to MpGABR via the THR228 amino acid residue. This site is highly conserved across 11 species in the Aphididae family of Hemiptera and may be a critical specific binding site for betulin in MpGABR in aphids. Mutagenesis and genome editing experiments revealed that betulin bound specifically to this amino acid residue in aphids but not in *Drosophila*, resulting in aphid death. This study elucidated the insecticidal mechanism of betulin involving the targeting of MpGABR, providing a sustainable green strategy for aphid control. However, given that betulin may affect a wider range of organisms, it should be used with caution.

## Results

### Control efficacy of betulin against aphids

Bioassays revealed that the LC$_{50}$ values of betulin and pymetrozine against *M. persicae* at 48 hr were 0.1641 and 1.0612 mg·mL$^{-1}$, respectively (*Figure 1—source data 1*). To assess the control efficacy of betulin, tobacco plants infected with *M. persicae* were exposed to betulin (0.1641 mg·mL$^{-1}$) and pymetrozine (1.0612 mg·mL$^{-1}$, as a positive control) in both the greenhouse and field. In the greenhouse test (*Figure 1A–D*), at 14 days after treatment, *M. persicae* in the CK group (negative control) reproduced rapidly, and tobacco leaves turned yellow and wrinkled, whereas tobacco leaves in the betulin and pymetrozine treatment groups were barely affected by the aphids (*Figure 1A and B*). The control efficacy of betulin against *M. persicae* was significantly greater than that of pymetrozine at 1 day after treatment (p<0.01), while there were no significant differences between the control efficacies of betulin and pymetrozine at 5, 9, or 14 days after treatment (*Figure 1E*). After 14 days of treatment, the control efficacies of betulin and pymetrozine reached 91.54% and 95.15%, respectively.

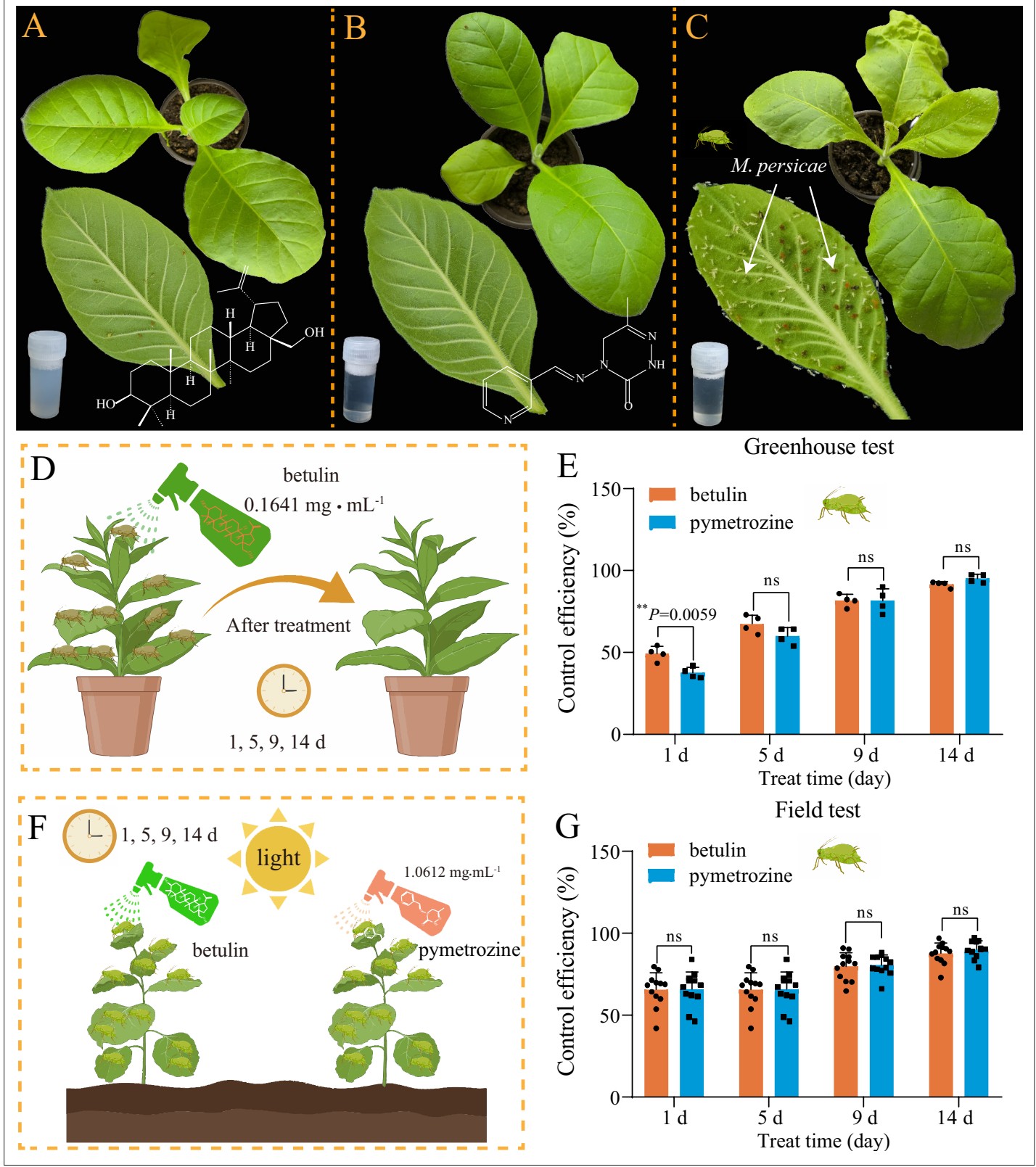

**Figure 1.** The aphicidal activity of plant-derived derivative betulin against *M. persicae.* (**A–G**) Representative images of the control effects of betulin (**A**, 0.1641 mg·mL⁻¹, 48 hr LC₅₀ value of betulin), pymetrozine (**B**, 1.0612 mg·mL⁻¹, a positive control), and CK (**C**, a negative control) against *M. persicae* in greenhouse (**A–E**) and field (**F, G**) tests after treatment for 14 days. The error bars represent standard deviation (SD) with n=4 (greenhouse test) and 12 (field test), respectively. **p<0.01. ns, not significant.

*Figure 1 continued on next page*

*Figure 1 continued*

The online version of this article includes the following source data for figure 1:

**Source data 1.** LC$_{50}$ values of betulin and pymetrozine against *M. persicae* at 48 hr, corresponding to *Figure 1*, panels D and F.

Throughout the field test (*Figure 1F*), no difference in the control efficacy was observed between betulin and pymetrozine, with values of 87.79% and 90.14%, respectively, after 14 days of treatment (*Figure 1G*). These results suggest that betulin has immense potential for development as a commercial aphid insecticide like pymetrozine.

## Expression patterns of genes in aphids after treatment with betulin

To investigate the expression patterns of genes in *M. persicae* after exposure to betulin, RNA-seq was performed on *M. persicae* with 0.1641 mg·mL$^{-1}$ betulin treatment for 48 hr and without (CK) (*Figure 2A*). Principal coordinate analysis (PCA) revealed that the expressed genes were strongly clustered in the CK group and betulin group (*Figure 2B*). Compared with those in the CK group, there were 130 up- and 41 downregulated significant differentially expressed genes (DEGs) in *M. persicae* after betulin treatment (*Figure 2C and D*, *Figure 2—source data 1*). To validate the RNA-seq results, 15 DEGs were randomly selected with different expression levels. The relative expression trends of the 15 DEGs determined via qRT-PCR were similar to those detected via RNA-seq, supporting the reliability of the RNA-seq data (*Figure 2E*). To further analyze the functions of the DEGs, Gene Ontology (GO) and Kyoto Encyclopedia of Genes and Genomes (KEGG) enrichment analyses were performed. GO term enrichment demonstrated that the DEGs were enriched mainly

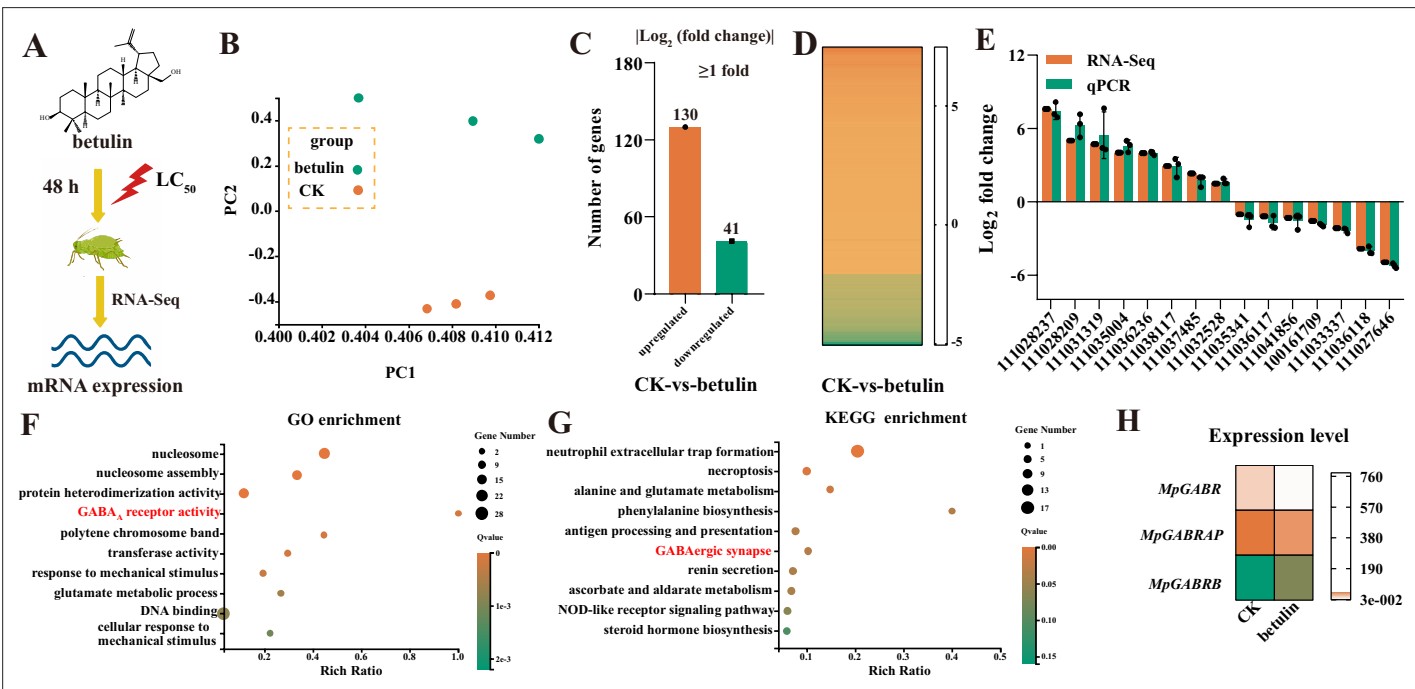

**Figure 2.** RNA-seq analysis revealed candidate targets for betulin against *M. persicae*. (**A**) Diagram of the candidate targets for betulin against aphids as determined by RNA-seq. Water containing 0.1% (vol/vol) Tween-80 and 3% (vol/vol) acetone was used as the control treatment (CK). (**B**) Principal coordinate analysis (PCA) of differentially expressed genes (DEGs) in RNA-seq. (**C–D**) Distribution (**C**) and heatmap (**D**) of the significantly DEGs in *M. persicae*. (**E**) qPCR validation of 15 DEGs identified by RNA-seq. The error bars represent SD with n=3. (**F–G**) Top 10 enriched Gene Ontology (GO) (**F**) and Kyoto Encyclopedia of Genes and Genomes (KEGG) (**G**) pathways of the DEGs. The 'rich ratio' was defined as the ratio of the number of DEGs enriched in the pathway to the total number of genes enriched in the same pathway. (**H**) Heatmap of the expression level of the GABR genes identified by RNA-seq. *MpGABR*, encoding GABA$_A$ receptor; *MpGABRAP*, encoding GABA$_A$ receptor-associated protein; *MpGABRB*, encoding GABA$_A$ receptor β subunit.

The online version of this article includes the following source data for figure 2:

**Source data 1.** Screening of genes significantly differentially expressed in *M. persicae* in response to betulin at 48 hr post treatment, corresponding to *Figure 2*, panel C.

in the GABA$_A$ receptor activity term (*Figure 2F*). Interestingly, KEGG enrichment also found that the DEGs were enriched in the GABAergic signaling-related pathway and GABAergic synapse terms (*Figure 2G*). Additionally, the expression of DEGs related to GABR, including *MpGABR* (encoding GABA$_A$ receptor), *MpGABRAP* (encoding GABA$_A$ receptor-associated protein), and *MpGABRB* (encoding GABA$_A$ receptor β subunit), in the betulin group was significantly lower than that in the CK group (*Figure 2H*). In particular, the log$_2$(fold change) value of *MpGABR* was the lowest, at –3.8 (*Figure 2—source data 1*). These results suggested that *MpGABR* may be a candidate target for betulin against *M. persicae*.

## *MpGABR* expression was inhibited by betulin

Analysis of the sequence information revealed that the MpGABR, MpGABRAP, and MpGABRB proteins have 694, 118, and 250 amino acids, respectively, with calculated molecular weights of 77.20, 14.08, and 28.01 kDa and isoelectric points of 9.85, 9.63, and 7.20, respectively (*Figure 3—source data 1*). Additionally, structure prediction indicated that MpGABR, MpGABRAP, and MpGABRB contain 4, 0, and 1 transmembrane helical domain, respectively (*Figure 3A*), implying that only MpGABR has a complete transmembrane structure.

To further verify the inhibitory effect of betulin on GABR-related gene expression, the relative expression of the three GABR-related genes was detected in *M. persicae* exposed to the LC$_{30}$, LC$_{50}$, and LC$_{70}$ of betulin for 48 hr. The relative expression of all three genes decreased gradually as the concentration of betulin increased (*Figure 3B–D*). After exposure to the LC$_{30}$ of betulin for 48 hr, the relative expression of *MpGABR*, *MpGABRAP,* and *MpGABRB* decreased by 82.91%, 10.53%, and 11.49%, respectively. These results indicated that *MpGABR* was the most sensitive to betulin. To further investigate the effect of betulin on the MpGABR protein, *M. persicae* was exposed to betulin concentrations of LC$_{30}$, LC$_{50}$, and LC$_{70}$ for 48 hr. As the concentration of betulin increased, the MpGABR protein content in *M. persicae* gradually decreased (*Figure 3E and F*). Compared with that in the CK group, the MpGABR protein content in *M. persicae* exposed to betulin concentrations of LC$_{30}$, LC$_{50}$, and LC$_{70}$ decreased by 33.68%, 44.89%, and 76.89%, respectively.

Furthermore, after these three genes were silenced via RNAi (*Figure 4D*), the expression levels of *MpGABR*, *MpGABRAP,* and *MpGABRB* in *M. persicae* significantly decreased by 64.57%, 63.28%, and 67.33%, respectively, compared with those in the control groups (DEPC-treated water and dsGFP) (p<0.0001, *Figure 4A–C*). After exposure to the LC$_{50}$ of betulin for 48 hr, the mortality of *M. persicae* with *MpGABR* silenced markedly increased (p<0.001) by 30.44%, compared with that in the control groups, whereas the mortalities of *M. persicae* in the *MpGABRAP*- and *MpGABRB*-silenced groups did not significantly differ from that in the control groups (*Figure 4E*). Additionally, western blotting analysis revealed that MpGABR protein expression significantly decreased by 69.91% after RNAi (p<0.0001, *Figure 4—figure supplement 1A and B*). These results further implied that *MpGABR* might be a target for betulin against *M. persicae*.

## Phylogenetic analysis of GABRs in insects

Furthermore, phylogenetic analysis of GABRs in insects was performed using the amino acid sequences of 70 GARB proteins from Hemiptera, Diptera, Lepidoptera, Thysanoptera, Hymenoptera, and Coleoptera. MpGABR (XP_022173711.1) was found to be genetically closely related to CAI6365831.1 from *Macrosiphum euphorbiae*, XP 008183008.2 from *Acyrthosiphon pisum*, and XP 060864885.1 from *Metopolophium dirhodum* (*Figure 5*, *Figure 5—source data 1*), indicating that MpGABR may be functionally analogous to CAI6365831.1, XP 008183008.2, and XP 060864885.1.

## Interaction between MpGABR and betulin

The WT (wild-type) MpGABR was expressed in *Escherichia coli* (*Figure 6A*). The binding affinity of betulin with WT MpGABR was further measured using MST. The $K_d$ value was 2.24 µM (*Figure 6B*), suggesting that betulin could bind to MpGABR. Additionally, voltage-clamp-based electrophysiological recording demonstrated that MpGABR strongly responded to betulin (*Figure 6C*). In the presence of only GABA, a fast inward current was generated. As the concentration of added betulin increased, the current and channel activity of MpGABR decreased correspondingly, with an EC$_{50}$ value of 20.66 µM (*Figure 6C and D*). These results indicated that betulin acted as an inhibitor of MpGABR.

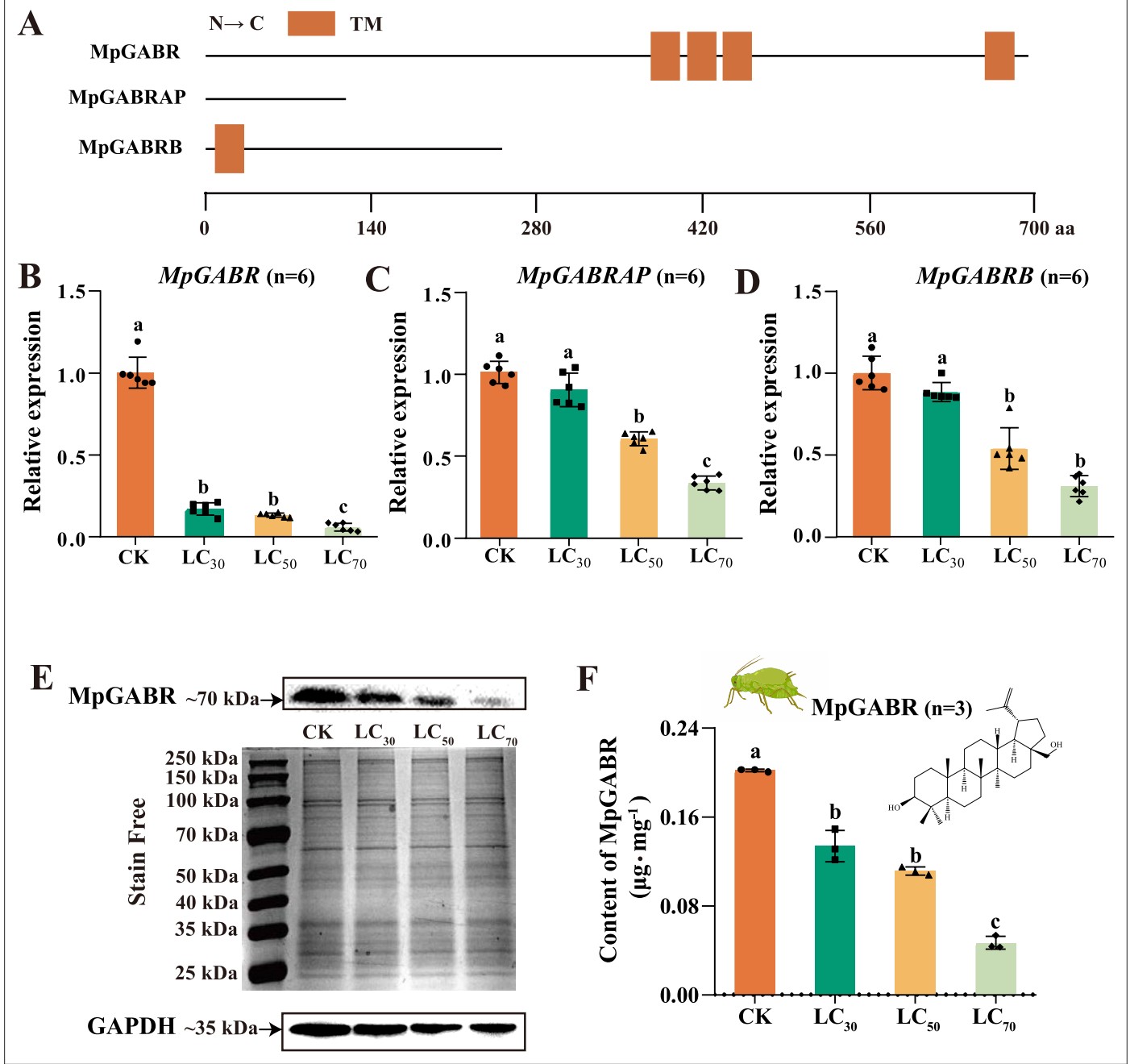

**Figure 3.** *MpGABR* expression in aphids was significantly inhibited by treatment with betulin. (**A**) Schematic drawing of MpGABR, MpGABRAP, and MpGABRB. TM, transmembrane helices. (**B–D**) qPCR expression analysis of *MpGABR* (**B**), *MpGABRAP* (**C**), and *MpGABRB* (**D**) transcripts in *M. persicae* exposed to LC$_{30}$, LC$_{50}$, and LC$_{70}$ of betulin for 48 hr. The bars represent the average (± SD). Different letters above the error bars indicate significant difference (analysis of variance [ANOVA], Tukey's test, p<0.05). (**E, F**) Western blotting analysis of MpGABR protein after betulin treatment for 48 hr at three different concentrations (LC$_{30}$, LC$_{50}$, and LC$_{70}$). The error bars represent SD. Different letters above the error bars indicate significant difference (ANOVA, Tukey's test, p<0.05).

The online version of this article includes the following source data for figure 3:

**Source data 1.** Complete sequence information for the *GABA$_A$ receptor* gene of *M. persicae* corresponding to *Figure 3*, panel A.

**Source data 2.** PDF file containing original membranes corresponding to *Figure 3*, panel E. GAPDH was used as a reference protein.

**Source data 3.** Original membranes without labels corresponding to *Figure 3*, panel E.

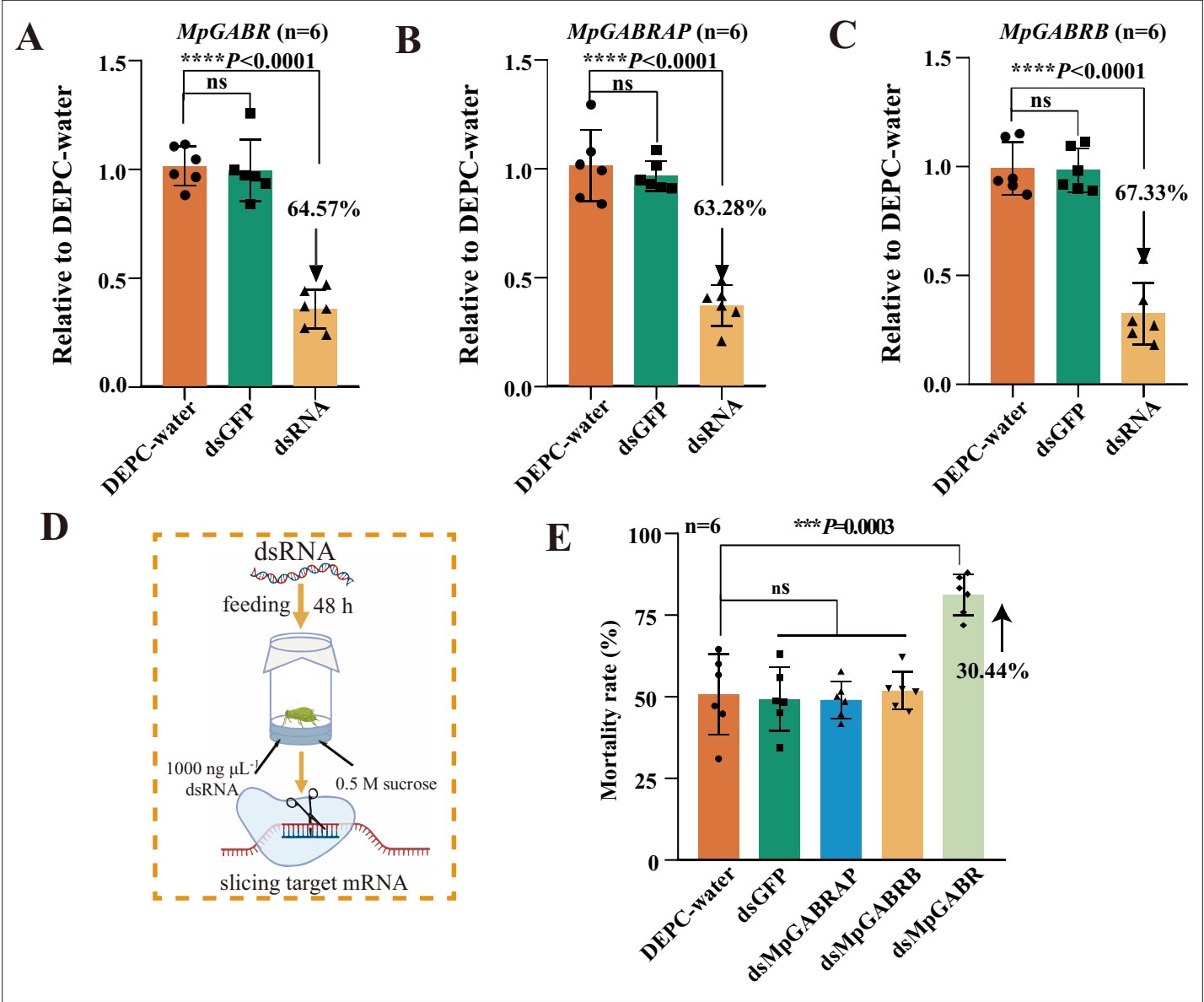

**Figure 4.** Silencing the expression of *MpGABR*, *MpGABRAP*, and *MpGABRB* via RNAi. (**A–C**) qPCR expression analysis of *MpGABR* (**A**), *MpGABRAP* (**B**), and *MpGABRB* (**C**) after RNAi at 48 hr post-dsRNA feeding relative to the expression levels after DEPC-water treatment. (**D**) Schematic drawing of the RNAi assay in *M. persicae*. (**E**) Mortality of aphids exposed to the $LC_{50}$ of betulin for 48 hr after RNAi. The error bars represent SD. An asterisk (*) on the error bar indicates a significant difference between the treatment and group CK according to t tests, ***p<0.001, ****p<0.0001. ns, not significant.

The online version of this article includes the following figure supplement(s) for figure 4:

**Figure supplement 1.** Protein expression of MpGABR after RNAi.

## Key sites for the binding of betulin to MpGABR

To further study the binding mode of betulin in the active pocket of MpGABR, molecular docking was performed to explore the structure-function relationships between betulin and MpGABR. The docking results revealed that the predicted binding energy between betulin and MpGABR was −6.38 kcal·mol$^{-1}$ (*Figure 7—source data 1*), verifying that betulin could act as a specific ligand for MpGABR. The three-dimensional binding pattern of betulin with MpGABR indicated that the four key amino acid residues (ARG224, ALA226, PHE227, and THR228) interacted with betulin in the MpGABR binding pocket (*Figure 7A*). Among the four key amino acid residues, ALA226 and THR228 interacted with betulin via hydrogen bonding in the MpGABR binding pocket (*Figure 7—source data 1*). Betulin generated nonconventional H-bonds (C…H) with ALA226 (3.31 Å) and conventional H-bonds (C-O

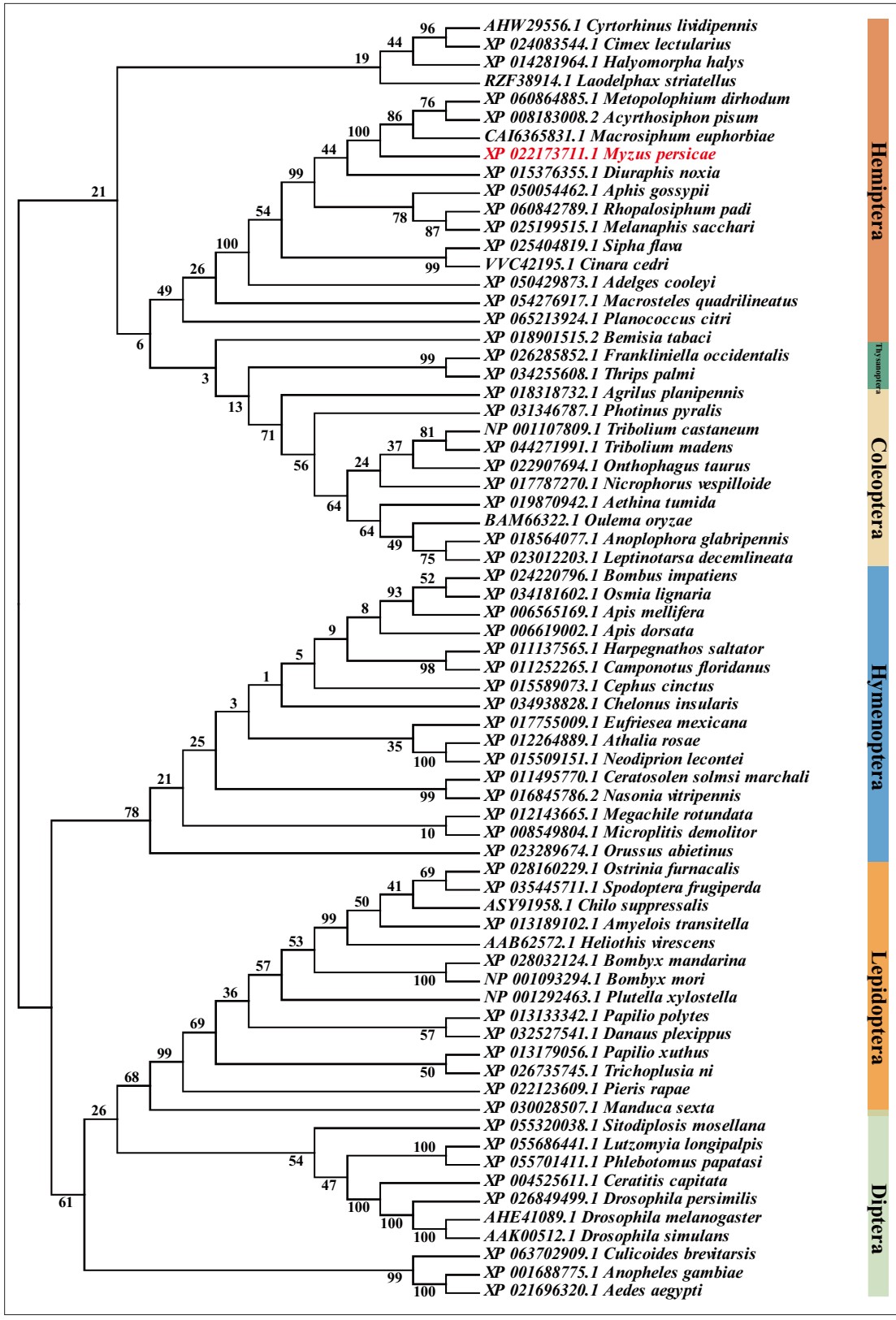

**Figure 5.** Phylogenetic analysis of GABR in insects. Neighbor-joining method with 1000 replicates based phylogenetic tree of GABR from six orders: Hemiptera, Diptera, Lepidoptera, Thysanoptera, Hymenoptera, and Coleoptera. Protein sequence alignment was performed using ClustalW in MEGA 7.

The online version of this article includes the following source data for figure 5:

**Source data 1.** Sequences and relevant information for phylogenetic analysis of GABA$_A$ receptor.

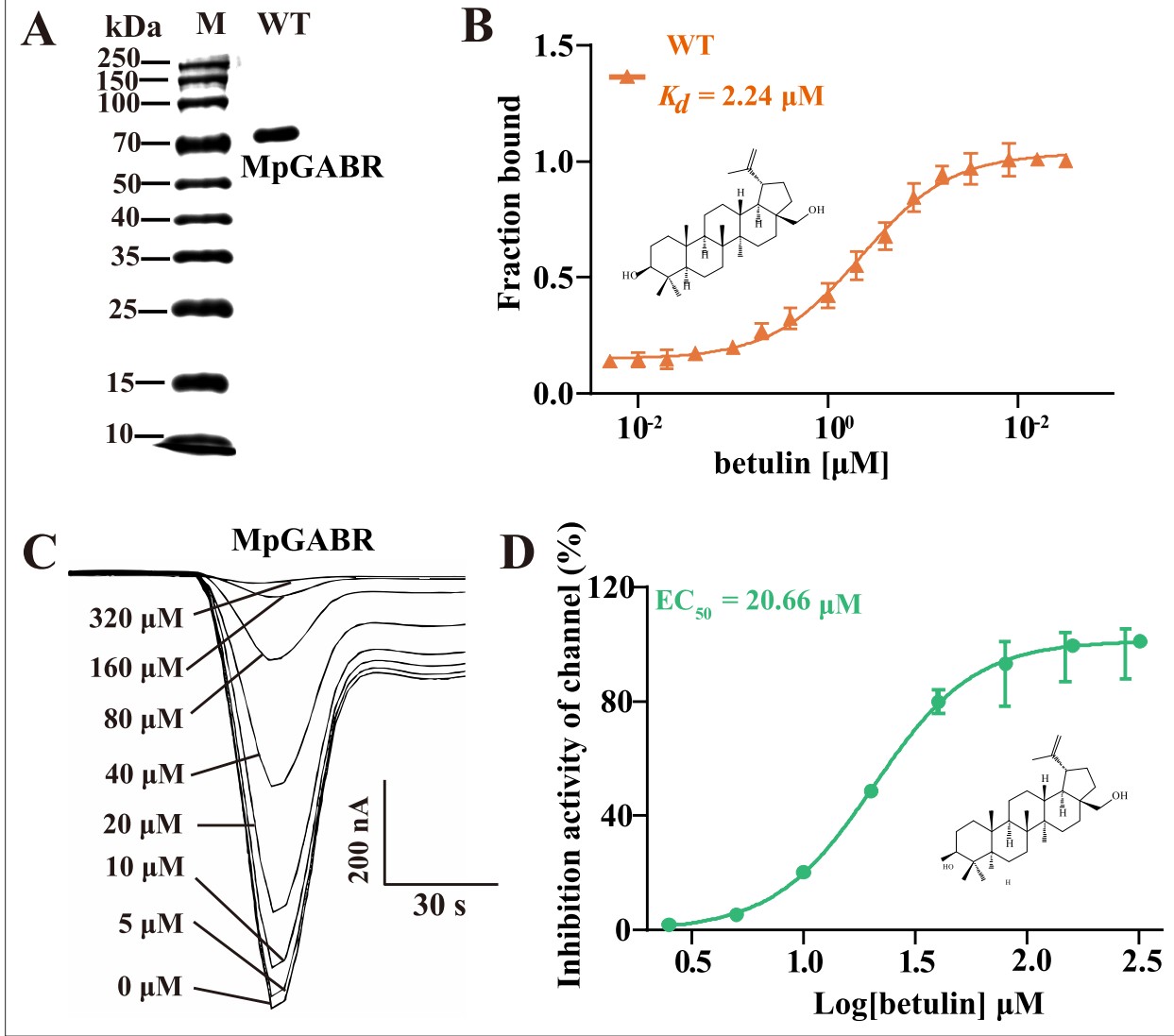

**Figure 6.** The interaction of betulin with MpGABR protein. (**A**) Expression of recombinant wild-type-aphid MpGABR (1, WT) protein by *E. coli*. (**B**) Quantification of the binding affinity of betulin with WT-aphid MpGABR using microscale thermophoresis (MST). Current responses (**C**) and inhibition activity (**D**) of MpGABR induced by different concentrations of betulin in the presence of GABA. 0 µM MpGABR indicates the presence of only GABA. The error bars represent SD with n=3.

The online version of this article includes the following source data for figure 6:

**Source data 1.** PDF file containing original gel corresponding to *Figure 6*, panel A.

**Source data 2.** Original gel without labels corresponding to *Figure 6*, panel A.

H) with THR228 (2.16 Å). Furthermore, sequence alignment of the key amino acid residues of GABR in different species from Hemiptera, Diptera, Lepidoptera, Hymenoptera, Thysanoptera, and Coleoptera indicated that, among the four residues, only THR228 was conserved across 11 species in the Aphididae family of Hemiptera (*Figure 7B*), implying that THR228 may be an essential specific site for the action of betulin on MpGABR in aphids. Mutants of aphid MpGABR were subsequently constructed, including R224A (ARG224 replaced by ALA), A226T (ALA226 replaced by THR), F227Y (PHE227 replaced by TYR, similar to the site in *Drosophila*), or T228R (THR228 replaced by ARG, similar to the site in *Drosophila*) (*Figure 7C*). The binding affinity of betulin with these mutants was measured using MST. The results revealed that the $K_d$ values of R224A, A226T, F227Y, T228R, and WT were 2.31, 2.27, 2.25, 5321, and 2.24 µM, respectively (*Figure 7D and E*, *Figure 7—source data 2*). This result suggested that the mutation at the THR228 site, with the highest $K_d$ value, markedly

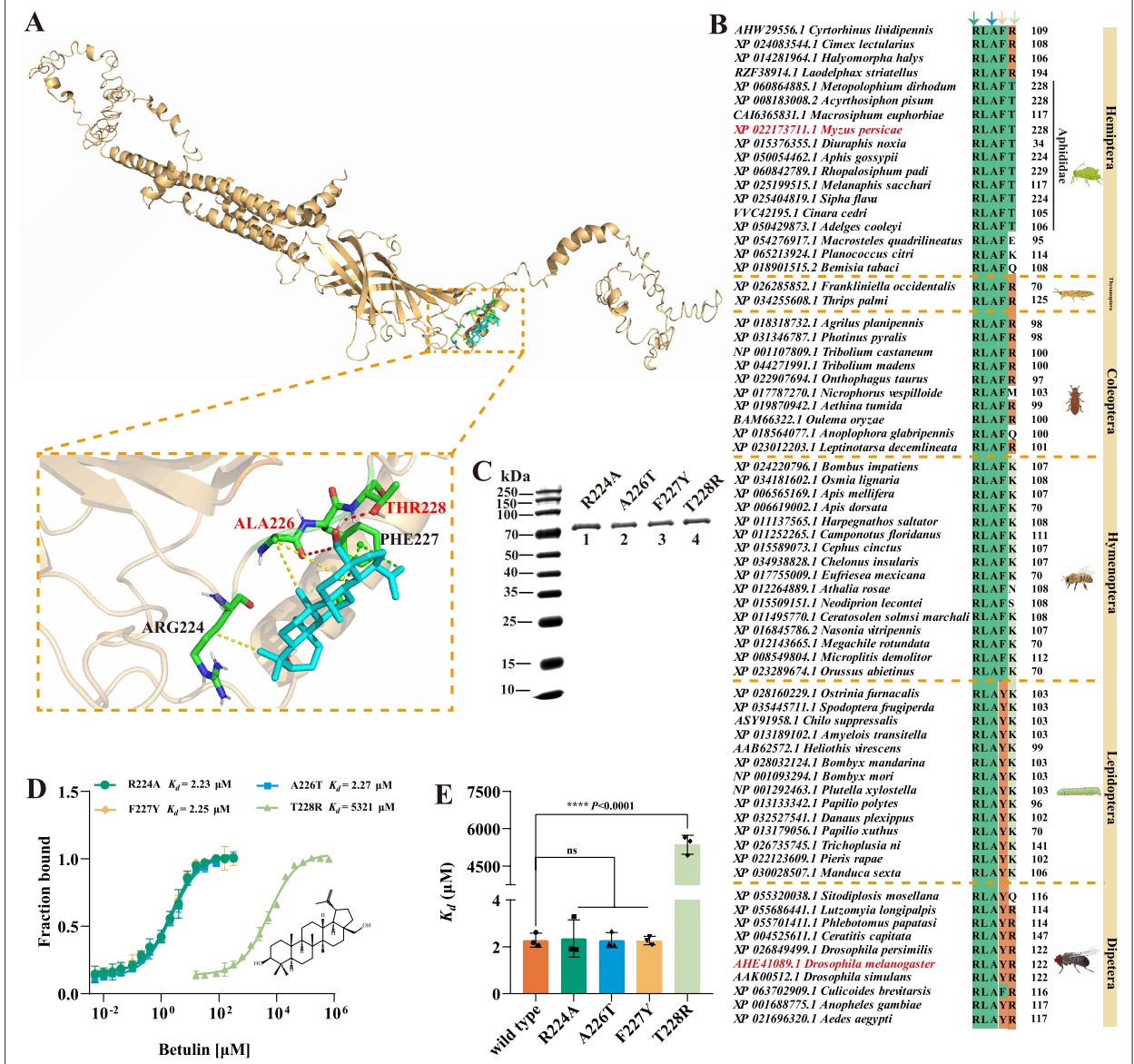

**Figure 7.** Molecular docking, binding site, and inhibitory effect of betulin on MpGABR. (**A**) Best conformations of betulin docked to the binding pocket of MpGABR in aphids. An enlarged view of the betulin binding sites in MpGABR is indicated by a dashed frame. Potential Pi-alkyl (green), alkyl (yellow), and hydrogen bond (red) interactions are indicated by dashed lines. (**B**) Sequence alignment of the key amino acids bound to betulin. The conserved residues among the different species in GABR are shown in orange, cyan, and light green. The numbers next to the amino acid indicate the site of the last residue of the key amino acids. (**C**) Expression of the recombinant mutation-type aphid MpGABR by *E. coli*. Lane 1: R224A, Lane 2: A226T, Lane 3: F227Y, Lane 4: T228R. (**D, E**) Quantification of the binding affinity of betulin with wild-type and mutant-type aphid MpGABR using microscale thermophoresis (MST). The error bars represent SD with n=3. * indicates a significant difference (Student's t test, ****p<0.0001).

The online version of this article includes the following source data and figure supplement(s) for figure 7:

**Source data 1.** Binding energy and nonbonding interactions between betulin and GABA$_A$ receptor, corresponding to *Figure 7*, panel A.

**Source data 2.** PDF file containing original gel corresponding to *Figure 7*, panel C.

**Source data 3.** Original gel without labels corresponding to *Figure 7*, panel C.

**Source data 4.** Parameters of dose-response curves from microscale thermophoresis experiments, corresponding to *Figure 7*, panel D.

**Figure supplement 1.** Betulin binding site on the domain of MpGABR.

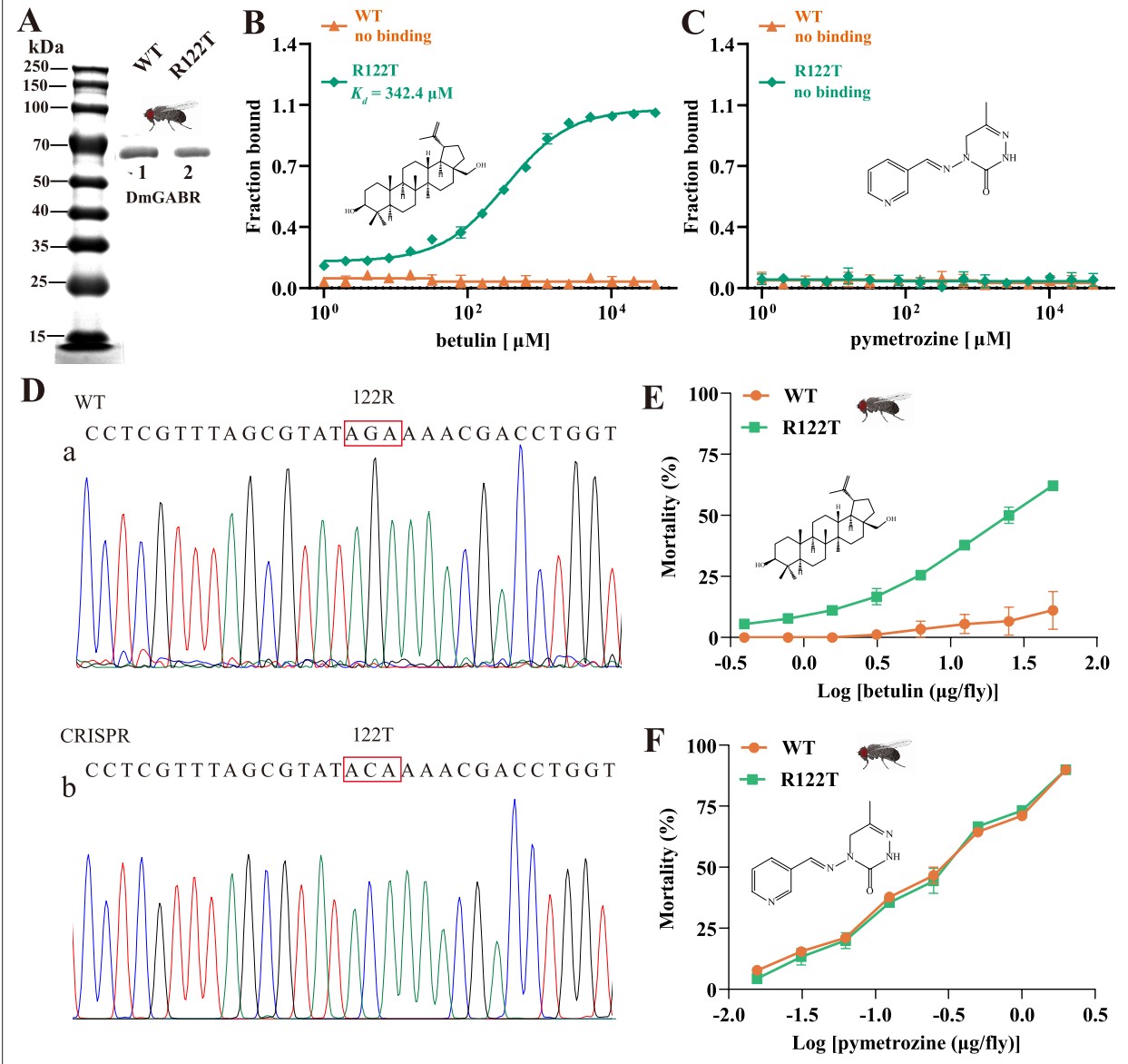

**Figure 8.** Gene editing in *Drosophila* validated the species-specific binding site of betulin. (**A**) Expression of the recombinant wild-type (WT) and mutation type (R122T) of DmGABR (*D. melanogaster*) by *E. coli*. Lane 1: WT, Lane 2: R122T. (**B, C**) Quantification of the binding affinity of betulin (**B**) and pymetrozine (**C**) with WT and R122T of DmGABR using microscale thermophoresis (MST). (**D**) Sanger sequencing of the *DmGABR* gene in flies. Direct sequencing chromatograms of PCR products amplified from a fragment of gDNA flanking the WT (**a**) and introduced R122T (**b**) mutant flies. (**E–F**) Toxicity curves of betulin (**E**) and pymetrozine (**F**) against WT and the DmGABR[R122T] of the Cas9 *Drosophila*. The error bars represent SD with n=3.

The online version of this article includes the following source data for figure 8:

**Source data 1.** PDF file containing original gel corresponding to *Figure 8*, panel A.

**Source data 2.** Original gel without labels corresponding to *Figure 8*, panel A.

**Source data 3.** LD$_{50}$ values of betulin and pymetrozine against *D. melanogaster* at 72 hr, corresponding to *Figure 8*, panels E and F.

reduced the binding ability of betulin to MpGABR, implying that the THR228 site may be an essential specific site for the binding of betulin to MpGABR.

## Betulin binds specifically to MpGABR via THR228

To further prove that THR228 is the specific binding site for betulin in MpGABR, the WT and mutant (R122T, equivalent to THR228 in MpGABR) *Drosophila* DmGABR proteins were expressed in *E. coli* and extracted (*Figure 8A and D*). The binding affinities of betulin and pymetrozine with DmGABR[WT]

and DmGABR$^{R122T}$ were evaluated using MST. As shown in *Figure 8B*, betulin was able to bind to DmGABR$^{R122T}$ ($K_d$ = 342.4 µM) but not to DmGABR$^{WT}$. However, pymetrozine exhibited no binding affinity for DmGABR$^{R122T}$ or DmGABR$^{WT}$ (*Figure 8C*). Moreover, to generate *Drosophila* mutant individuals carrying DmGABR$^{R122T}$, the CRISPR/Cas9 genomic editing strategy was used. After exposure to different concentrations of betulin, the mortality rate of DmGABR$^{R122T}$ *Drosophila* was significantly greater than that of DmGABR$^{WT}$ *Drosophila* (*Figure 8E*). In addition, the LD$_{50}$ value of betulin against DmGABR$^{R122T}$ *Drosophila* at 72 hr was 27.19 µg·fly$^{-1}$, whereas the LD$_{50}$ value of betulin against DmGABR$^{WT}$ *Drosophila* could not be calculated because the lethal concentration exceeded 1000 µg·fly$^{-1}$ (*Figure 8—source data 1*). Additionally, there was no significant difference in mortality between DmGABR$^{WT}$ and DmGABR$^{R122T}$ *Drosophila* after exposure to pymetrozine (*Figure 8F*). The LD$_{50}$ values of pymetrozine against DmGABR$^{WT}$ and DmGABR$^{R122T}$ *Drosophila* at 72 hr were 0.23 and 0.24 µg·fly$^{-1}$, respectively (*Figure 8—source data 1*). Moreover, THR228 is located at the neurotransmitter-gated ion-channel ligand-binding domain in MpGABR (*Figure 7—figure supplement 1*), implying that betulin may play a role as a competitive antagonist of MpGABR. Taken together, these findings suggested that betulin bound specifically to MpGABR via THR228, resulting in the death of *M. persicae*.

## Discussion

Although chemical pesticides are effective in pest control, the long-term unreasonable application of these substances has led to the emergence of resistant pests, environmental deterioration, and deleterious effects on nontarget organisms, including humans, raising widespread and intense concerns (*Devi et al., 2022*). These challenges have made it urgent for the development of alternative strategies for pest control. Recently, the utilization of plant secondary metabolites as insecticides has become increasingly popular as an eco-friendly and biocontrol approach (*Ling et al., 2022*; *Ayilara et al., 2023*). Our previous study revealed that *P. davidiana*, a wild relative of cultivated peach, strongly resists *M. persicae* by accumulating high contents of betulin (*Wang et al., 2022a*; *Wang et al., 2024*). In addition, betulin, a lupane-type triterpene, possesses potent insecticidal activity and is a promising substance for the development of novel insecticides for aphid control. In this study, the insecticidal effect of betulin was further evaluated by comparing the control efficacy of betulin with that of pymetrozine against aphids in greenhouses and fields (*Figure 1A–G*). These results indicate that betulin has a similar control effect to pymetrozine and has immense potential for development as a plant-derived insecticide.

Terpenes are a diverse group of plant secondary metabolites that can increase the resistance of plants to insect herbivores through direct (*Wang et al., 2025a*) and indirect (*Wang et al., 2025b*) defense mechanisms. In direct defense against herbivores, triterpenes play important roles in diverse biological activities, including antiparasitic, insecticidal, and antifeedant activities (*Tian et al., 2021*; *Kuzminac et al., 2023*). Azadirachtin, a tetracyclic triterpenoid compound isolated from the Indian neem tree (*Azadirachta indica*), is one of the most prominent commercial biopesticides, exhibiting strong insect antifeedant properties, as well as growth- and reproduction-regulating effects (*Dawkar et al., 2019*; *Bae et al., 2022*). Besides, triterpene glycoside compounds play crucial roles in the defense of tobacco (*Nicotiana attenuata*) against tobacco hornworm (*Manduca sexta*) larvae (*Yang et al., 2024*). Although betulin exhibits various pharmacological activities (*Amiri et al., 2020*; *Li et al., 2022*; *Yan et al., 2022*; *Lv et al., 2023*), reports on its insecticidal activity are limited. Encouragingly, the effects of betulin and its derivatives on pests have attracted increasing attention. Betulinic acid and its derivatives showed larvicidal activity against *A. aegypti* larvae (*da Silva et al., 2016*). Betulin-cinnamic acid-related hybrid compound 5b exhibited strong aphicidal activity, and compound 2l could destroy the ultrastructure of midgut cells and significantly inhibit the activity of α-amylase in diamondback moth (*Pl. xylostella* L.) larvae (Huang et al.). Our previous studies also indicated that betulin possesses potent insecticidal activity and is a key endogenous secondary metabolite related to the defense of peach against *M. persicae* (*Wang et al., 2022a*). Elucidating the insecticidal mechanism of betulin against aphids will provide a basis for the development of novel aphicides and sustainable strategies for aphid control.

GABA receptors have been confirmed to be targets of terpenoids that impair insect neuronal function in herbivores (*Guo et al., 2023*). GABRs are heteropentameric ligand-gated ion channels in the central nervous system that conduct chloride and bicarbonate ions. These receptors are targets of numerous drugs for the treatment of neuropsychiatric disorders (*Thompson, 2024*). A variety of

terpenoids act as positive allosteric modulators or NCAs of GABRs, such as diterpenoids (isopimaric acid and miltirone), sesquiterpenoids (picrotoxin, bilobalide, and ginkgolides), and monoterpenoids (α-thujone and thymol) (*Guo et al., 2023*). In this study, both GO and KEGG enrichment analyses revealed that the DEGs identified by RNA-seq were enriched in GABAergic signaling-related pathways (*Figure 2F and G*). Additionally, the expression of the DEGs related to GABRs, particularly *MpGABR* (*Figure 2—source data 1*), in the betulin group was significantly lower than that in the CK group (*Figure 2H*). Besides, the relative expression of *MpGABR*, *MpGABRAP*, and *MpGABRB* decreased gradually after *M. persicae* was exposed to the $LC_{30}$, $LC_{50}$, and $LC_{70}$ of betulin for 48 hr (*Figure 3B–D*). Among them, *MpGABR* was the most sensitive to betulin, and its expression was reduced by 82.91% after exposure to the $LC_{30}$ of betulin for 48 hr. Furthermore, compared with the control group, the *M. persicae* group with *MpGABR* silenced by RNAi presented a significant increase in mortality (p<0.001), by 30.44%, after 48 hr of exposure to the $LC_{50}$ of betulin (*Figure 4E*). Collectively, these results suggest that betulin may have insecticidal effects on aphids by inhibiting *MpGABR* expression. The regulation of gene expression is sophisticated and delicate (*Pope and Medzhitov, 2018*). The regulatory network controlling *GABR* expression remains unclear. In adult rats, epileptic seizures have been reported to increase the levels of brain-derived neurotrophic factor, which in turn prompted the transcription factors CREB and ICER to reduce the gene expression of the GABR α1 subunit (*Lund et al., 2008*). In *Drosophila*, it has been demonstrated that WIDE AWAKE, which regulated the onset of sleep, interacted with the GABR and upregulated its expression level (*Liu et al., 2014*). In the *Drosophila* brain, circular RNA circ_sxc was found to inhibit the expression of miR-87-3-p in the brain through sponge adsorption, thereby regulating the expression of neurotransmitter receptor ligand proteins, including GABR, and ensuring the normal function of synaptic signal transmission in brain neurons (*Li et al., 2024*). However, it remains unclear how betulin reduces *MpGABR* expression, and further research is needed.

Additionally, betulin has been reported to be able to bind to GABRs (*Manayi et al., 2016*). We further investigated the interaction of betulin with the MpGABR protein. The MST assay revealed that betulin was able to bind to MpGABR ($K_d$ = 2.24 μM) (*Figure 6B*), which is consistent with previous findings showing that betulin binds to GABA receptors in mouse brains in vitro (*Muceniece et al., 2008*). Voltage-clamp-based electrophysiological recordings indicated that betulin acted as an inhibitor ($EC_{50}$=20.66 μM) for MpGABR (*Figure 6C and D*). Subsequent molecular docking analysis suggested that four key amino acid residues (ARG224, ALA226, PHE227, and THR228) interact with betulin in the MpGABR binding pocket (*Figure 7A*), among which merely ALA226 and THR228 interact with betulin via hydrogen bonding (*Figure 7—source data 1*). The results of the sequence alignment revealed that only THR228 was conserved across 11 species in the Aphididae family of Hemiptera (*Figure 7B*). Furthermore, evaluation of the ability of betulin to bind to MpGABR mutants with mutations at those four sites revealed that the ability of betulin to bind to T228R was significantly weaker than its ability to bind to the WT (*Figure 7D and E*, *Figure 7—source data 2*), indicating that THR228 is an essential specific site for the binding of betulin to MpGABR. Moreover, to further prove that THR228 is the specific binding site for betulin in MpGABR, the binding affinities of betulin with the WT and mutant (R122T, equivalent to THR228 in MpGABR) *Drosophila* DmGABR proteins were assessed using MST. The results showed that betulin was able to bind to DmGABR$^{R122T}$ ($K_d$ = 342.4 μM) but not DmGABR$^{WT}$ (*Figure 8B*). Additionally, after exposure to different concentrations of betulin, the mortality rate of DmGABR$^{R122T}$ *Drosophila* was significantly greater than that of DmGABR$^{WT}$ *Drosophila* (*Figure 8E*, *Figure 8—source data 1*). Similarly, a previous study indicated that the R122G amino acid site substitution, generated via RNA editing, affects the sensitivity of *Drosophila* to fipronil (*Es-Salah et al., 2008*). Together, these findings suggest that betulin binds specifically to MpGABR via THR228, acting as an inhibitor of MpGABR and causing aphid death. Studies on key amino acids that are crucial for GABR function have primarily focused on transmembrane regions. For instance, based on the mutational research and *Drosophila* GABR modeling approach, multiple key amino acids have been identified as insecticide targets in the transmembrane domain (*Nakao and Banba, 2021*). *Guo et al., 2023* proposed that amino acid substitutions in transmembrane domain 2 contribute to terpenoid insensitivity during plant-insect coevolution. However, these studies have neglected the extracellular domain. Our study signified that betulin targets the THR228 site in the extracellular domain of MpGABR, which is conserved only in the Aphididae family. Therefore, betulin is speculated to be a specific insecticidal substance evolved by plants in response

to aphid infestation. Besides, further verification is needed to determine whether betulin is toxic to other insect species.

GABRs belong to the cysteine loop (Cys)-loop superfamily of neurotransmitter receptors and are essential heteropentameric ligand-gated ion channels in the central nervous system. After GABA binds to the extracellular Cys-loop of a GABR (*Ashby et al., 2012*), the GABR is activated and opens up its central pore to allow chloride ions to pass through, thereby hyperpolarizing neurons and attenuating excitatory neurotransmission (*Tremblay et al., 2016*). In insects, GABRs play important roles in circadian rhythms (*Schellinger et al., 2022*), sleep (*Chaturvedi et al., 2022*), movement (*Eick et al., 2022*), and olfactory memory (*Yamagata et al., 2021*). Additionally, GABR is a critical target for a variety of insecticides and ectoparasiticides. Insecticides targeting GABRs are categorized into NCAs and competitive antagonists (CAs) according to their different binding sites. NCAs block chloride channels in nerve cells by interacting with amino acid residues of the GABA-gated chloride channel, causing a conformational change in the receptor, which interferes with the normal function of the central nervous system and ultimately leads to insect death (*Nakao and Banba, 2021*; *Guo et al., 2023*). A series of first- and second-generation NCA insecticides have been successfully developed. Picrotoxinin, isolated from *Anamirta cocculus* fruit, is the oldest natural NCA and is toxic to houseflies (*Tong et al., 2023*). Among the first-generation NCAs, polychlorocycloalkanes, including dieldrin and lindane, were commonly used as pesticides in the middle and late 20th centuries (*Hainzl et al., 1998*; *Tanaka, 2019*). As second-generation NCAs, fipronil and its derivatives are commercially available phenylpyrazole insecticides that have been widely used for agricultural pest control (*Sheng et al., 2018*; *Li et al., 2021*). Additionally, CAs bind to orthogonal binding sites, such as the GABA recognition site in the extracellular region, competitively inhibiting the binding of GABA to its receptor, leading to toxic effects in insects. Given the lack of interference from existing insect resistance mechanisms, CAs hold promise for the development of efficient new insecticides. Novel 1,6-dihydro-6-iminopyridazine-derived insecticides, as CAs, have insecticidal properties against common cutworms and houseflies (*Liu et al., 2022*). Besides, nootkatone, a sesquiterpenoid, also acts as a CA to induce insect mortality (*Norris et al., 2022*). Fluralaner, an isooxazoline ectoparasiticide, inhibits both parasites and *A. aegypti* by acting as a CA (*Wang et al., 2022b*; *Asahi et al., 2023*). Our results revealed that the binding site (THR228) for betulin in MpGABR was located in the extracellular neurotransmitter-gated ion-channel ligand-binding domain (*Figure 7—figure supplement 1*), implying that betulin acts as a CA of MpGABR. Although the mechanism by which betulin competes with GABA for binding to MpGABR requires further experimental validation, our work may have provided a novel target for developing insecticides. In this study, betulin, on the one hand, inhibited the expression of *MpGABR* and, on the other hand, specifically bound to MpGABR through THR228, acting as an inhibitor of MpGABR and causing aphid death (*Figure 9*).

The development of bioinsecticides should not only focus on the toxic effects of active substance on target organisms, but also on their influence on the ecosystem (*Haddi et al., 2020*). Although our results indicate that betulin has specific toxicity to aphids, previous studies have reported that betulin and its derivatives had effects on *P. xylostella* L. (*Huang et al., 2025*), *A. aegypti* (*de Almeida Teles et al., 2024*), and *D. melanogaster* (*Lee and Min, 2024*). Therefore, further research is needed to determine whether there are other insecticidal mechanisms or off-target effects of betulin. Additionally, betulin exhibits a wide range of pharmacological activities (*Amiri et al., 2020*), which have been used to treat various diseases, such as cancer (*Lv et al., 2023*), glioblastoma (*Li et al., 2022*), inflammation (*Szlasa et al., 2023*), and hyperlipidemia (*Tang et al., 2011*). Before applying betulin in the field, it is necessary to fully verify and consider whether betulin has any impact on farmers' health. Furthermore, will betulin cause residue or diffusion during the field application process? Will long-term application promote the evolution of resistance to aphids or other insects? These issues also need further experimental verification. In summary, before any field application, further research on the environmental behavior, degradation process, and safety of betulin is needed.

## Conclusion

Betulin, a key metabolite in the aphid-resistant wild peach *P. davidiana*, possesses potent aphicidal effects on *M. persicae*. This study confirmed that betulin exhibited excellent control efficacy against *M. persicae* in both greenhouse and field experiments. RNA-seq, qRT-PCR, and western blotting assays revealed that betulin significantly inhibited the expression of *MpGABR* in aphids. In addition,

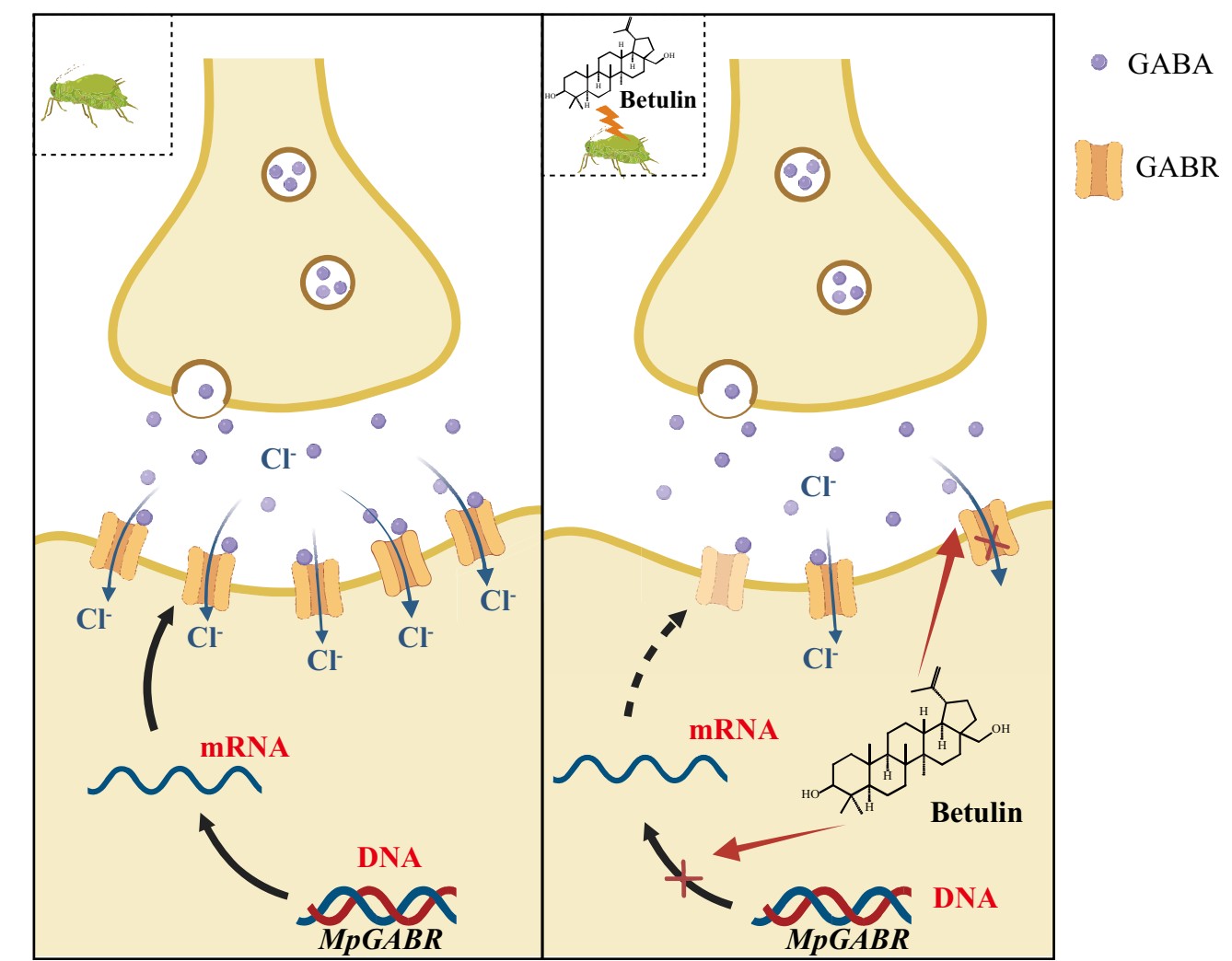

**Figure 9.** Proposed model for the mechanism of action against *M. persicae* by targeting GABA$_A$ receptors (GABR). After exposure to betulin, the expression of *MpGABR* was inhibited, and the level of MpGABR protein decreased, resulting in a decrease in the channel of chloride ion influx. Besides, betulin directly and specifically binds to the amino acid residue THR228 of MpGABR, thereby disabling it.

RNAi-mediated silencing of *MpGABR* markedly increased the sensitivity of aphids to betulin. Moreover, MST and voltage-clamp-based electrophysiological recording assays indicated that betulin was able to bind to MpGABR ($K_d$ = 2.24 µM) and acted as an inhibitor (EC$_{50}$=20.66 µM) of MpGABR. Molecular docking analyses suggested that the amino acid residue THR228, which is highly conserved across 11 species in the Aphididae family of Hemiptera, might be a critical specific binding site for betulin in MpGABR. Mutagenesis and genome editing assays revealed that betulin bound specifically to this amino acid residue in aphids but not in *Drosophila*, resulting in aphid death. Collectively, the results suggest that the aphicidal effects of betulin on aphids occur in a two-pronged manner: on the one hand, betulin inhibits *MpGABR* expression; on the other hand, it specifically binds to MpGABR via THR228 and acts as an inhibitor of MpGABR. Elucidating the insecticidal mechanism of betulin against aphids will provide a basis for the development of novel insecticides and sustainable strategies for aphid control.

# Materials and methods

## Test insects

*M. persicae* populations were collected from tobacco (*Nicotiana tabacum* L.) leaves. WT flies (*D. melanogaster*) were purchased from Fungene Biotechnology Co., Ltd., Jiangsu, China. The test insects were cultured in a greenhouse (25°C, 70% relative humidity, and 14 hr light/10 hr dark cycle) in Beibei, Chongqing, China.

## Leaf bioassay

The toxicity of betulin against *M. persicae* was measured using a slip-dip bioassay as previously described (*Wang et al., 2022a*). Thirty apterous adult *M. persicae* individuals were gently transferred onto tobacco (*N. tabacum*) leaves using a fine brush. The tobacco leaves were subsequently immersed in 0.1% (vol/vol) Tween-80 and 3% (vol/vol) acetone solution containing betulin or pymetrozine (as a positive control) at 7 doses (0, 0.0625, 0.125, 0.25, 0.5, 1, and 2 mg·mL$^{-1}$) for 5 s. After drying, the leaves with *M. persicae* were placed on wet filter paper and cultured on a water-soaked sponge for 48 hr. Three replicates were tested for each treatment. Furthermore, the $LC_{30}$, $LC_{50}$, and $LC_{70}$ values of betulin were calculated using log-probit analysis via IBM SPSS Statistics (v.22.0, Chicago, IL, USA) (*Zhou et al., 2024b*).

## Greenhouse and field bioassays

The control effects of betulin against *M. persicae* in the greenhouse and field were evaluated by a spraying-based method as described previously (*Zhou et al., 2023*). The commercial insecticide pymetrozine and an aqueous solution containing 0.1% (vol/vol) Tween-80 and 3% (vol/vol) acetone were employed as positive and negative controls, respectively. Sprays of 0.1641 mg·mL$^{-1}$ ($LC_{50}$) betulin and 1.0612 mg·mL$^{-1}$ ($LC_{50}$) pymetrozine (*Figure 1—source data 1*) were prepared in aqueous solutions, including 3% (vol/vol) acetone and 0.1% (vol/vol) Tween-80, respectively. Before the control effect assays, each tobacco (*N. tabacum* L.) seedling was pre-inoculated with 60 apterous adults of *M. persicae*. In the greenhouse trial, each treatment included eight replications, and each replication consisted of three tobacco plants growing under controlled conditions as described above. The field trial (Beibei District, Chongqing, China) was conducted under natural conditions with an average temperature of 20°C. Each treatment had eight replications, covering an area of 20 m$^2$, with three protected rows per replication. Furthermore, each tobacco plant was sprayed with 20 mL of the corresponding solution with an electric sprayer. Mortality was recorded at 1, 5, 9, and 14 days after treatment. The control efficacy of betulin against *M. persicae* was calculated according to the methods of *Zhou et al., 2023*.

## RNA-seq assay

After exposure to 0.1641 mg·mL$^{-1}$ (48 hr $LC_{50}$) betulin for 48 hr, 100 apterous *M. persicae* adults were sampled for each of three replicates, and total RNA was extracted via TRIzol reagent (Tiangen, Beijing, China) to construct mRNA sequencing libraries. RNA-seq analysis was performed on the Ion Proton BGISEQ-500 platform (Beijing Genomic Institute, BGI, China). After the raw reads were filtered, the clean reads were aligned to the *M. persicae* reference genome, which was downloaded from the National Center for Biotechnology Information database, using HISAT2 (v2.1.0) (*Kim et al., 2015*). The expression levels of genes in each sample were normalized to the fragments per kilobase million mapped reads value. The DEGs between groups were identified using the fold change factor ($|Log_2(FC)|>1$). Functional enrichment analysis of DEGs was performed using the GO and KEGG databases.

## Validation by qRT-PCR

The gene-specific primers used for qRT-PCR, which were designed with Primer Premier 5.0, are listed in *Supplementary file 1* - Table S1. According to the manufacturer's specifications for the Bio-Rad iQ SYBR Green Supermix Kit, qRT-PCR was performed with the following conditions: 60°C for 15 s, followed by 48 cycles of 60°C for 30 s, and 72°C for 30 s. The *ribosomal protein S18* gene (*RPS18*) was used as the reference gene to normalize the expression levels. The relative expression levels of

the target genes were calculated using the $2^{-\Delta\Delta Ct}$ approach (**Zhou et al., 2024b**). Three biological replicates were used for each assay.

## Cloning, bioinformatics, and phylogenetic analysis

The cDNA of *M. persicae* was synthesized from the extracted total RNA using the PrimeScript 1st Strand cDNA Synthesis Kit (TaKaRa, Japan). The coding sequences of *MpGABR* (encoding GABA$_A$ receptor), *MpGABRAP* (encoding GABA$_A$ receptor-associated protein), and *MpGABRB* (encoding GABA$_A$ receptor β subunit) were cloned from *M. persicae* cDNA using the corresponding specific primers (**Supplementary file 1**—Table S1) and then inserted separately into the pGEM-T Easy vector. The recombinant plasmids were subsequently transferred into *E. coli* DH5α competent cells for sequencing. BioXM (v2.7.1) (http://202.195.246.60/BioXM/) was used to analyze the sequence information of these three genes, including the length, molecular weight, and isoelectric point of the deduced amino acid sequence. MEGA7 was used for multiple amino acid sequence alignment (ClustalW) and phylogenetic analysis (neighbor-joining method with 1000 replicates) of MpGABR.

## RNAi assay

To knock down the target genes (*MpGABR*, *MpGABRAP*, and *MpGABRB*), *MpGABR-*, *MpGABRAP-*, and *MpGABRB*-dsRNA were artificially synthesized using the dsRNA Synthesis Kit (Thermo Scientific, Vilnius, Lithuania, EU). In accordance with a previously described method (**Zhou et al., 2024a**), fresh tobacco leaves were cut into 3.0-cm-diameter discs and placed in an oven at 50°C for 3 min. Subsequently, the dried leaf discs were exposed to 10 µL of diethyl pyrocarbonate-treated nuclease-free water (DEPC-water), dsGFP (green fluorescent protein, 1000 ng·µL$^{-1}$, negative control), and *MpGABR-*, *MpGABRAP-*, or *MpGABRB*-dsRNA (1000 ng·µL$^{-1}$) at 25°C for 5 hr. After the leaf discs had absorbed the solution, they were placed on wet filter paper on a water-soaked sponge. Subsequently, 30 apterous adult *M. persicae*, starved for 24 hr, were fed on the treated leaf discs for 48 hr. The surviving aphids were collected after exposure for qRT-PCR. Three biological replicates were included for each treatment.

## Prokaryotic expression of *MpGABR* and western blotting

To produce the MpGABR protein in vitro, an *E. coli* expression system was constructed (**Zhou et al., 2021**). Briefly, the coding sequence of *MpGABR* was inserted into the expression vector PET-B2M, obtained from BGI (Shenzhen, China), and transformed into BL21 (*DE3*) competent cells. To induce the expression of the MpGABR protein, 2 mM isopropyl β-D-1-thiogalactopyranoside (IPTG) was used, and then, the transformed BL21 cells were incubated at 180 rpm and 30°C for 24 hr. Finally, the protein was extracted and purified using Solarbio GST-tag Purification Resin for subsequent assays.

The purified MpGABR protein resolved on a 15% SDS-PAGE (sodium dodecyl sulfate-polyacrylamide gel electrophoresis) gel and then transferred to a polyvinylidene fluoride (PVDF) membrane at 20 V for 1 hr. Moreover, 1×TBST (Tris-Borate-Sodium Tween-20) containing 5% fat-free milk powder was used to block the PVDF membrane. Then, the PVDF membrane was incubated with an antibody against MpGABR (polyclonal rabbit antibody, provided by GeneCreate Biotechnology) at 25°C for 1 hr. Subsequently, the membrane was further incubated with the corresponding horseradish peroxidase (HRP)-conjugated goat anti-rabbit immunoglobulin G antibody at 25°C for 1 hr and then washed three times with TBST. Finally, the bands on the membrane were visualized using a Chemi Doc MP Imaging System (Bio-Rad) after incubation with a chemiluminescence reagent (ECL, Bio-Rad). GAPDH was used as a reference protein.

## Molecular docking

In accordance with a previously reported method (**Zhou et al., 2023**), the protein structure and function of MpGABR were predicted via the I-TASSER server (Iterative Threading ASSEmbly Refinement). AutoDockTools software (v1.5.7) was used to conduct molecular docking. Furthermore, PyMOL (v3.1) and Discovery Studio (v4.5) were used to evaluate the molecular docking model.

## Site-directed mutagenesis

The coding sequence of *MpGABR* was inserted into the PET-B2M vector and used as the template for site-directed mutagenesis. In accordance with the manufacturer's specifications, the Fast MultiSite

Mutagenesis Kit (TransGen, Beijing, China) was used to generate the multisite mutations in MpGABR (*Ma et al., 2020*). Then, the MpGABR mutants, with the mutations of R224A (ARG224 replaced by ALA), A226T (ALA226 replaced by THR), F227Y (PHE227 replaced by TYR), or T228R (THR228 replaced by ARG), were amplified, and the amplicons were used to replace the *Bam*HI-*Sma*I fragment in the recombinant MpGABR plasmid.

## Method for *Drosophila* genome editing

To generate *Drosophila* mutants bearing the DmGABR$^{R122T}$ mutation, homologous to THR228 in MpGABR, the Clustered Regularly Interspaced Short Palindromic Repeats-Cas9 (CRISPR/Cas9) genomic editing strategy was used. First, CRISPR Optimal Target Finder (http://targetfinder.flycrispr. neuro.brown.edu/) was used to generate short guide RNA (sgRNA) targets without off-target hits. Then, the Precision gRNA Synthesis Kit (Thermo Scientific) was used to synthesize the sgRNA. The 228 bp homologous arm in R122T, for homology-directed repair, was subsequently employed to synthesize single-stranded DNA (ssDNA) bearing the desired mutation (DmGABR$^{R122T}$). Subsequently, 1 nL of a mixture containing 160 ng·μL$^{-1}$ ssDNA, 300 ng·μL$^{-1}$ both sgRNAs, and 90 ng·μL$^{-1}$ Cas9 protein was injected into *Drosophila* embryos via the Drummond Nanoject III (Broomall, PA, USA) system. CRISPR/Cas9-induced mutations were verified by amplifying and sequencing genomic DNA flanking the genome editing target site. Finally, according to the traditional topical application and bioassay approach described previously (*Zhou et al., 2023*), the acute insect toxicity of betulin in genome-modified or nonmodified (WT) 3-day-old adult flies was determined. The oligonucleotide sequences used for PCR, the sgRNA synthesis template, and the fragment of donor DNA used for homology-directed repair are listed in *Supplementary file 1* - Table S2.

## MST assay

The binding affinity of MpGABR for betulin was measured via a NanoTemper Technologies Mono-lith Kit (MO, Munich, Germany) according to the manufacturer's specifications. Briefly, MpGABR was fluorescently labeled via a protein labeling kit (RED-NHS 2nd generation, Monolith) and then mixed with 16 concentrations (from 5 nM to 60 mM) of betulin. The mixtures were subsequently loaded into NanoTemper capillaries for MST measurements with 40% LED and 60% MST power at 25°C. MO Control NanoTemper Technologies GmbH software (v2.3) and MO Affinity Analysis software (v2.3, NanoTemper Technologies GmbH) were used to acquire and analyze the recorded data, respectively. Three replicates were set for each concentration. The dissociation constant ($K_d$), indicating the binding affinity, was calculated using data from three replicates for each experiment.

## Electrophysiological recording

In accordance with previously described methods (*Chen et al., 2019*), oocytes isolated from mature healthy *Xenopus* females were digested at 25°C for 1 hr to remove the follicular layer. Individual oocytes were selected to express *MpGABR* and incubated at 18°C with 1×Ringer's solution (96 mM NaCl, 5 mM MgCl$_2$, 5 mM HEPES, 2 mM KCl, and 0.8 mM CaCl$_2$) supplemented with 550 mg·mL$^{-1}$ sodium pyruvate, 100 mg·mL$^{-1}$ streptomycin, 50 mg·mL$^{-1}$ tetracycline, and 5% dialyzed horse serum. After 3 days of incubation, the oocytes were used for electrophysiological recording. GABA was dissolved in 1×Ringer's solution to prepare 100 μM GABA solution. Subsequently, the 100 μM GABA solutions containing different concentrations of betulin (0, 5, 10, 20, 40, 80, 160, 320 μM) were used to perfuse the oocytes. The whole-cell currents were recorded via a two-electrode voltage-clamp. The data were collected and analyzed using Digidata 1440A and pCLAMP 10.2 software (Axon Instruments Inc, Union City, CA, USA), respectively (*Xu et al., 2020*).

## Statistical analysis

Statistical analysis was performed using SPSS (v22). Statistical significance was determined using one-way analysis of variance (ANOVA) for differences among groups. The replicate numbers are indicated by the black dots in the figures.

## Acknowledgements

This research was supported by the National Natural Science Foundation of China (32402492, 32302394), Special Funding for Chongqing Postdoctoral Research Project (2312013543265057),

Sichuan Science and Technology Program (2025ZNSFSC1120), and Fundamental Research Fund for the Central Universities of China (SWU-KR25018).

## Additional information

### Funding

| Funder | Grant reference number | Author |
|---|---|---|
| National Natural Science Foundation of China | 32402492 | Junxiu Wang |
| National Natural Science Foundation of China | 32302394 | Hong Zhou |
| Special Funding for Chongqing Postdoctoral Research Project | 2312013543265057 | Junxiu Wang |
| Sichuan Science and Technology Program | 2025ZNSFSC1120 | Hong Zhou |
| Fundamental Research Fund for the Central Universities of China | SWU-KR25018 | Hong Zhou |

The funders had no role in study design, data collection and interpretation, or the decision to submit the work for publication.

### Author contributions

Junxiu Wang, Data curation, Funding acquisition, Visualization, Writing – original draft, Writing – review and editing; Matthana Klakong, Data curation, Validation, Investigation, Methodology; Qiuyu Zhu, Investigation, Methodology; Jinting Pan, Yudie Duan, Methodology; Lirong Wang, Yong Li, Resources; Jiangbo Dang, Danlong Jing, Writing – review and editing; Hong Zhou, Data curation, Supervision, Funding acquisition, Visualization, Writing – original draft, Writing – review and editing

### Author ORCIDs

Junxiu Wang (iD) https://orcid.org/0000-0002-6453-3477
Hong Zhou (iD) https://orcid.org/0000-0003-2438-622X

Reviewer #1 (Public review): https://doi.org/10.7554/eLife.107598.3.sa1
Reviewer #2 (Public review): https://doi.org/10.7554/eLife.107598.3.sa2
Author response https://doi.org/10.7554/eLife.107598.3.sa3

## Additional files

### Supplementary files

Supplementary file 1. Primer sequences.

MDAR checklist

### Data availability

RNA sequence data were deposited in NCBI (accession number PRJNA1334570). All data generated or analyzed during this study are included in the manuscript and supporting files.

The following dataset was generated:

| Author(s) | Year | Dataset title | Dataset URL | Database and Identifier |
|---|---|---|---|---|
| Wang J | 2025 | Transcriptome analyses of aphids treated with betulin | https://www.ncbi.nlm.nih.gov/bioproject/PRJNA1334570 | NCBI BioProject, PRJNA1334570 |

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
