## [Editor Report · eLife Assessment]

This **important** study identifies a plant-derived metabolite, betulin, as an effective natural insecticide against aphids and uncovers its specific molecular target. The evidence is **compelling**, combining greenhouse and field efficacy trials with rigorous molecular, genetic, and electrophysiological approaches that converge on a conserved binding site in the aphid GABA receptor. While additional work is needed to fully assess potential off-target effects and ecological safety, the study provides a strong mechanistic foundation. These findings will be of interest to researchers in plant biology, chemical ecology, and sustainable pest management.

---

## [Referee Report · Reviewer #1 (Public review)]

Wang, Junxiu et al. investigated the underlying molecular mechanisms of the insecticidal activity of betulin against the peach aphid, Myzus persicae. There are two important findings described in this manuscript: (a) betulin inhibits the gene expression of GABA receptor in the aphid, and (b) betulin binds to the GABA receptor protein, acting as an inhibitor. The first finding is supported by RNA-Seq and RNAi, and the second one is convinced with MST and electrophysiological assays. Further investigations on the betulin binding site on the receptor protein provided a fundamental discovery that T228 is the key amino acid residue for its affinity, thereby acting as an inhibitor, backed up by site-directed mutagenesis of the heterologously-expressed receptor in *E. coli* and by CRISPR-genome editing in *Drosophila*.

Comments on revisions:

All of my review comments have been addressed, and the manuscript has been revised accordingly.

---

## [Referee Report · Reviewer #2 (Public review)]

Summary:

This important study shows that betulin from wild peach trees disrupts neural signaling in aphids by targeting a conserved site in the insect GABA receptor. The authors present a nicely integrated set of molecular, physiological, and genetic experiments to establish the compound's species-specific mode of action. While the mechanistic evidence is solid, the manuscript would benefit from a broader discussion of evolutionary conservation and potential off-target ecological effects.

Strengths:

The main strengths of the study lie in its mechanistic clarity and experimental rigor. The identification of a betulin-binding single threonine residue was supported by (1) site-directed mutagenesis and (2) functional assays. These experiments strongly support the specificity of action. Furthermore, the use of comparative analyses between aphids and fruit flies demonstrates an important effort to explore species specificity, and the integration of quantitative data further enhances the robustness of the conclusions.

Comments on revisions:

The revision satisfactorily addresses my concerns on evolutionary context, methodological clarity, and ecological risk.

---

## [Author Response]

The following is the authors’ response to the original reviews.

**Reviewer #1 (Public review):**
Wang, Junxiu et al. investigated the underlying molecular mechanisms of the insecticidal activity of betulin against the peach aphid, Myzus persicae. There are two important findings described in this manuscript: (a) betulin inhibits the gene expression of GABA receptor in the aphid, and (b) betulin binds to the GABA receptor protein, acting as an inhibitor. The first finding is supported by RNA-Seq and RNAi, and the second one is convinced with MST and electrophysiological assays. Further investigations on the betulin binding site on the receptor protein provided a fundamental discovery that T228 is the key amino acid residue for its affinity, thereby acting as an inhibitor, backed up by site-directed mutagenesis of the heterologously-expressed receptor in *E. coli* and by CRISPR-genome editing in *Drosophila*.Although the manuscript does have strengths in principle, the weaknesses do exist: the manuscript would benefit from more comprehensive analyses to fully support its key claims in the manuscript. In particular:(1) The Western blotting results in Figure 5A & B appear to support the claim that betulin inhibits GABR gene expression (L26), as a decrease in target protein levels is often indicative of suppressed gene expression. The result description for Figure 5A & B is found in L312-L316, within Section 3.6 ("Responses of MpGABR to betulin"), where MST and voltage-clamp assays are also presented. It seems the observed decrease in MpGABR protein content is due to gene downregulation, rather than a direct receptor protein-betulin interaction. However, this interpretation lacks discussion or analysis in either the corresponding results section or the Discussion. In contrast, Figures 5C-F are specifically designed to illustrate protein-betulin interactions. Presenting Figure 5A & B alongside these panels might lead to confusion, as they support distinct claims (gene expression vs. protein binding/inhibition). Therefore, I recommend moving Figure 5A & B either to the end of Figure 3 or to a separate figure altogether to improve clarity and logical flow. A minor point in the Western blotting experiment is that although GAPDH was used as a reference protein, there is no explanation in the corresponding M&M section.

We thank the reviewer for the concise and accurate summary and appreciate the constructive feedback on the article’s strengths and weaknesses.

(A) According to your suggestion, the original Figure 5A and B have been inserted into Figure 3, following Figure 3D. The original Figure 3E-I has been saved as a new figure, to illustrate the RNAi assay.

(b) “GAPDH was used as a reference protein” has been supplied in the M&M section, see

Line 209.

(2) The description of the electrophysiological recording experiment is unclear regarding the use of GABA. I didn't realize that GABA, the true ligand of the GABA receptor, was used in this inhibition experiment until I reached the Results section (L321), which states, "In the presence of only GABA, a fast inward current was generated." Crucially, no details are provided on the experiment itself, including how GABA was applied (e.g., concentration, duration, whether GABA was treated, followed by betulin, or vice versa). This information is essential for reproducibility. Please ensure these details are thoroughly described in the corresponding M&M section.

We thank the reviewer for the valuable comments.

(a) Detailed information on how to apply GABA has been added to the corresponding M&M section (Lines 260-263): After 3 days of incubation, the oocytes were used for electrophysiological recording. GABA was dissolved in 1 × Ringer's solution to prepare 100 µM GABA solution. Subsequently, the 100 µM GABA solutions containing different concentrations of betulin (0, 5, 10, 20, 40, 80, 160, 320 µM) were used to perfuse the oocytes.

(b) Additionally, we also checked other contents of M&M section to ensure that sufficient detail has been supplied.

(3) The phylogenetic analysis, particularly concerning Figures 4 and 6B, needs significant attention for clarity and representativeness. First, your claim that MpGABR is only closely related to CAI6365831.1 (L305-L310) is inconsistent with the provided phylogenetic tree, which shows MpGABR as equally close to Metopolophium dirhodum (XP_060864885.1) and Acyrthosiphon pisum (XP_008183008.2). Therefore, singling out only Macrosiphum euphorbiae (CAI6365831.1) is not supported by the data. Second, the representation of various insect orders is insufficient. All 11 sequences in the Hemiptera category (in both Figure 4 and Figure 6B) are exclusively from the Aphididae family. This small subset cannot represent the highly diverse Order Hemiptera. Consequently, statements like "only THR228 was conserved in Hemiptera" (L338), "The results of the sequence alignment revealed that only THR228 was conserved in Hemiptera" (L430), or "THR228... is highly conserved in Hemiptera" (L486) are not adequately supported. Third, similar concerns apply to the Diptera order, which includes 10 Drosophila and 2 mosquito samples (not diverse or representative enough), and likely to other orders as well. Thereby, the Figure 6B alignment should be revised accordingly to reflect a more accurate representation or to clarify the scope of the analysis. Fourth, there's a discrepancy in the phylogenetic method used: the M&M section (L156) states that MEGA7, ClustalW, and the neighbor-joining method were used, while the Figure 4 caption mentions that MEGA X, MUSCLE, and the Maximum likelihood method were employed. This inconsistency needs to be clarified and made consistent throughout the manuscript. Fifth, I have significant concerns about the phylogenetic tree itself (Figure 4). A small glitch was observed at the Danaus plexippus node, which raises suspicion regarding potential manipulation after tree construction. More critically, the tree, especially within Coleoptera, does not appear to be clearly resolved. I am highly concerned about whether all included sequences are true GABR orthologs or if the dataset includes partial or related sequences that could distort the phylogeny. Finally, for Figure 6B, both protein (XP_) and nucleotide (XM_) sequences were mix used. I recommend using the protein sequences instead of nucleotide sequences in this figure panel, as protein sequences are more directly informative.

We thank the reviewer for the careful reading and valuable comments.

(a) Firstly, according to your comments, phylogenetic analysis has been re-performed with more represent species from each Order (Fig. 5 and Fig. 7B). The results revealed that only THR228 was conserved across 11 species in the Aphididae family of Hemiptera. Therefore, the expressions like "only THR228 was conserved in Hemiptera" have been revised to “among the four residues, only THR228 was conserved across 11 species in the Aphididae family of Hemiptera” (Line 106, Line 369, Line 477, and Lines 563-564).

(b) We have modified the description of Fig. 5 (the original Fig. 4): MpGABR (XP_022173711.1) was found to be genetically closely related to CAI6365831.1 from Macrosiphum euphorbiae, XP 008183008.2 from Acyrthosiphon pisum, and XP 060864885.1 from Metopolophium dirhodum (Fig. 5 and Table S6). See Lines 342-346.

(c) Phylogenetic analysis was performed using MEGA7 with multiple amino acid sequence alignment (ClustalW) and the neighbor-joining method. We have revised the Fig. 5 (the original Fig. 4) caption to make it accurate and consistent throughout the manuscript.

(d) We are sorry about the small glitch at the Danaus plexippus node. Actually, after the phylogenetic tree was constructed, it was imported in Adobe Illustration for coloring and classification annotation. There may have been operational errors during the process of resizing the image, resulting in the occurrence of the small glitch. Besides, the unclear clustering of Coleoptera may be due to improper regulation of distance (pixels) of branch from nodes. Again, thanks for your careful reading. We have rebuilt the phylogenetic tree.

(e) Based on your suggestion, the sequence IDs have been unified as the protein sequence IDs (Fig. 5, Fig. 7B and Table S6)

(4) The Discussion section requires significant revision to provide a more insightful and interpretative analysis of the results. Currently, much of the section primarily restates findings rather than offering deeper discussion. For instance, L409-L419 restate the results, followed by the short sentence "Collectively, these results suggest that betulin may have insecticidal effects on aphids by inhibiting MpGABR expression". It could be further expanded to make it beneficial to elaborate on proposed mechanisms by which gene expression might be suppressed, including any potential transcription factors involved. In contrast, while L422-L442 also initially summarize results, the subsequent paragraph (L445-L472) effectively discusses the potential mechanisms of inhibitory action and how mortality is triggered, which is a good model for other parts of the section. However, all the discussion ends up with a short statement, "implying that betulin acts as a CA of MpGABR" (L472), which appears to be a leap. The inference that betulin acts as a competitive antagonist (CA) is solely based on the location of its extracellular binding site, which does not exactly overlap with the GABA binding site. It needs stronger justification or actually requires further experimental validation. The authors should consider rephrasing this statement to acknowledge the need for additional studies to definitively confirm this mechanism of action.

We appreciate the reviewer's careful reading and valuable feedback, which will certainly enhance the quality of our manuscript.

(a) Possible reasons for the effect of betulin on MpGABR expression have been discussed in our manuscript (Lines 455-466): The regulation of gene expression is sophisticated and delicate (Pope and Medzhitov 2018). The regulatory network controlling GABR expression remains unclear. In adult rats, epileptic seizures has been reported to increase the levels of brain-derived neurotrophic factor (BDNF), which in turn prompted the transcription factors CREB and ICER to reduce the gene expression of the GABR α1 subunit (Lund et al. 2008). In Drosophila, it has been demonstrated that WIDE AWAKE, which regulated the onset of sleep, interacted with the GABR and upregulated its expression level (Liu et al. 2014). In Drosophila brain, circular RNA circ_sxc was found to inhibit the expression of miR-87-3p in the brain through sponge adsorption, thereby regulating the expression of neurotransmitter receptor ligand proteins, including GABR, and ensuring the normal function of synaptic signal transmission in brain neurons (Li et al. 2024). However, it remains unclear how betulin reduces the expression of MpGABR, and further research is needed.

(b) In the Discussion section, we acknowledged the need for further research to ultimately confirm the mechanism by which betulin competes with GABA for binding to MpGABR (Lines 532-535): Although the mechanism by which betulin competes with GABA for binding to MpGABR requires further experimental validation, our work may have provided a novel target for developing insecticides.

(c) Besides, we have added the discussion of the sensitivity of GABA receptor to betulin in Discussion section (Lines 491-501): Studies on key amino acids that are crucial for GABR function has primarily focused on transmembrane regions. For instance, based on the mutational research and Drosophila GABR modeling approach, multiple key amino acids were identified as insecticide targets in the transmembrane domain (Nakao and Banba 2021). Guo et al. proposed that amino acid substitutions in the transmembrane domain 2 contribute to terpenoid insensitivity during plant-insect coevolution (Guo et al. 2023). However, these studies have neglected the extracellular domain. Our study signified that betulin targets the THR228 site in the extracellular domain of MpGABR, which is conserved only in the Aphididae family. Therefore, betulin is speculated to be a specific insecticidal substance evolved by plants in response to aphid infestation. Besides, further verification is needed to determine whether betulin is toxic to other insect species.

(d) Furthermore, the discussion of potential ecological risks of deploying betulin as a bioinsecticide has been elaborated in our manuscript (Lines 538-553): The development of bioinsecticides should not only focus on the toxic effects of active substance on target organisms, but also on their influence on the ecosystem (Haddi et al. 2020). Although our results indicate that betulin has specific toxicity to aphids, previous studies have reported that betulin and its derivatives had effects on Plutella xylostella L. (Huang et al. 2025), Aedes aegypti (de Almeida Teles et al. 2024), and *Drosophila melanogaster* (Lee and Min 2024). Therefore, further research is needed to determine whether there are other insecticidal mechanisms or off target effects of betulin. Additionally, betulin exhibits a wide range of pharmacological activities (Amiri et al. 2020), which have been used to treat various diseases, such as cancer (Lv 2023), glioblastoma (Li et al. 2022), inflammation (Szlasa et al. 2023) and hyperlipidemia (Tang et al. 2011). Before applying betulin in the field, it is necessary to fully verify and consider whether betulin has any impact on farmers' health. Furthermore, will betulin cause residue or diffusion in the process of field application? Will long-term application promote the evolution of resistance to aphids or other insects? These issues also need further experimental verification. In summary, before any field application, further research is needed on the environmental behavior, degradation process, and safety of betulin.

**Reviewer #2 (Public review):**
Summary:This important study shows that betulin from wild peach trees disrupts neural signaling in aphids by targeting a conserved site in the insect GABA receptor. The authors present a nicely integrated set of molecular, physiological, and genetic experiments to establish the compound's species-specific mode of action. While the mechanistic evidence is solid, the manuscript would benefit from a broader discussion of evolutionary conservation andpotential off-target ecological effects.Strengths:The main strengths of the study lie in its mechanistic clarity and experimental rigor. The identification of a betulin-binding single threonine residue was supported by (1) site-directed mutagenesis and (2) functional assays. These experiments strongly support the specificity of action. Furthermore, the use of comparative analyses between aphids and fruit flies demonstrates an important effort to explore species specificity, and the integration of quantitative data further enhances the robustness of the conclusions.Weaknesses:There are several important limitations that need to be addressed. The manuscript does not explore whether the observed sensitivity to betulin reflects a broadly conserved feature of GABA receptors across animal lineages or a more lineage-specific adaptation. This evolutionary context is crucial for understanding the broader significance of the findings.In addition, while the compound's aphicidal effect is well established, the potential for off-target effects in non-target organisms - especially vertebrates - remains unaddressed, despite prior evidence that betulin interacts with mammalian GABAa receptors. There is little discussion on the ecological or environmental safety of exogenous betulin application, such as persistence, degradation, or exposure risks.

We sincerely thank the reviewer for the time and effort dedicated to our manuscript's detailed review and assessment. The revision suggestions were constructive, and we have provided a point-by-point response to address them.

(a) Briefly introduce the evolutionary conservation of GABA receptors has been added in the Introduction (Lines 90-98): Previous study has proposed that vertebrate and human GABR genes maintain a broad and conservative gene clustering pattern, while in invertebrates, this pattern is missing, indicating that these gene clusters formed early in vertebrate evolution and were established after diverging from invertebrates. Notably, invertebrates each possess a unique GABR gene pair, which are homologous with human GABR α and β subunits, suggesting that the existing GABR gene cluster evolved from an ancestral α - β subunit gene pair (Tsang et al. 2006). During the coevolution of plants and insects, the duplications and amino acid substitutions in GABR may be beneficial for the adaptation to insecticides and terpenoid compounds (Guo et al. 2023).

(b) We have added the discussion of the sensitivity of GABA receptor to betulin in Discussion section (Lines 491-501): Studies on key amino acids that are crucial for GABR function has primarily focused on transmembrane regions. For instance, based on the mutational research and Drosophila GABR modeling approach, multiple key amino acids were identified as insecticide targets in the transmembrane domain (Nakao and Banba 2021). Guo et al. proposed that amino acid substitutions in the transmembrane domain 2 contribute to terpenoid insensitivity during plant-insect coevolution (Guo et al. 2023). However, these studies have neglected the extracellular domain. Our study signified that betulin targets the THR228 site in the extracellular domain of MpGABR, which is conserved only in the Aphididae family. Therefore, betulin is speculated to be a specific insecticidal substance evolved by plants in response to aphid infestation. Besides, further verification is needed to determine whether betulin is toxic to other insect species.

(c) The discussion of potential ecological risks of deploying betulin as a bioinsecticide has been elaborated in our manuscript (Lines 538-553): The development of bioinsecticides should not only focus on the toxic effects of active substance on target organisms, but also on their influence on the ecosystem (Haddi et al. 2020). Although our results indicate that betulin has specific toxicity to aphids, previous studies have reported that betulin and its derivatives had effects on Plutella xylostella L. (Huang et al. 2025), Aedes aegypti (de Almeida Teles et al. 2024), and *Drosophila melanogaster* (Lee and Min 2024). Therefore, further research is needed to determine whether there are other insecticidal mechanisms or off target effects of betulin. Additionally, betulin exhibits a wide range of pharmacological activities (Amiri et al. 2020), which have been used to treat various diseases, such as cancer (Lv 2023), glioblastoma (Li et al. 2022), inflammation (Szlasa et al. 2023) and hyperlipidemia (Tang et al. 2011). Before applying betulin in the field, it is necessary to fully verify and consider whether betulin has any impact on farmers' health. Furthermore, will betulin cause residue or diffusion in the process of field application? Will long-term application promote the evolution of resistance to aphids or other insects? These issues also need further experimental verification. In summary, before any field application, further research is needed on the environmental behavior, degradation process, and safety of betulin.

**Reviewer #1 (Recommendations for the authors):**
(1) L28 Provide the full name of MST.

Thanks for your suggestion. The full name of MST, microscale thermophoresis, has been supplied.

(2) L87 in the Order Hemiptera.

Thanks for your suggestion. Corrected.

(3) L99 "Leaf bioassay" would be better to differentiate the greenhouse and field bioassays.

Thanks for your suggestion. Corrected.

(4) L104 It should be 7 doses, including the "0 mg/mL" control.

Thanks for your suggestion. Corrected.

(5) L104 Since the LC50 of pymetrozine is 1.0612 mg/mL, a wider range of doses should have been tested compared to the dose range of betulin.

Thanks for your comment.

(a) Firstly, seven doses (0, 0.0625, 0.125, 0.25, 0.5, 1, and 2 mg.mL^-1^) were set to calculate the LC50 of betulin and pymetrozine. Since the LC50 values of betulin and pymetrozine are 0.1641 and 1.0612 mg.mL^–1^, respectively, which are within the set range, indicating that the set dose range is reasonable and the LC50 values of betulin and pymetrozine are reliable.

(b) To compare the control effects of betulin and pymetrozine against M. persicae, LC50 of betulin (0.1641 mg.mL^-1^) and pymetrozine (1.0612 mg.mL^-1^) were used to treat M. persicae.

(6) L109 Greenhouse and field bioassays.

Thanks for your suggestion. Corrected.

(7) L112 Tween-80 and acetone in L103. Keep the order consistent throughout the manuscript.

Thanks for your suggestion. Corrected.

(8) L122 Mortality was recorded at 1, 5, 9, and 14 days after treatment. Revise the other similar mistakes throughout the manuscript (e.g. L250, L254, L255, L256, L259, etc.).

Thanks for your suggestion. Corrected.

(9) L126 apterous instead of wingless (keep a consistent expression).

Thanks for your suggestion. Corrected.

(10) L138 Primer Premier?

Thanks for your comment. Corrected.

(11) L141 Add RPS18 primers in Table S2.

Thanks for your comment. Corrected.

(12) L155 MEGA7 vs. MEGAX (as described in the Figure 4 caption).

Thanks for your comment. Corrected.

(13) L156 NJ method vs. ML method (as described in the Figure 4 caption).

Thanks for your comment. Corrected.

(14) L157 2.7. RNAi assay (Remove "In vitro" and re-number the following M&M sections accordingly).

Thanks for your comment. Corrected.

(15) L163 Add dsGFP primers in Table S2.

Thanks for your comment. Corrected.

(16) L166 apterous instead of wingless (keep a consistent expression).

Thanks for your comment. Corrected.

(17) L172 Add the source of pET-B2M vector.

pET-B2M vector was obtained from BGI (Shenzhen, China), which has been added in our manuscript (Line 194).

(18) L195 coding sequence instead of cDNA.

Thanks for your comment. Corrected.

(19) L198 the mutations of R224A ...

Thanks for your comment. Corrected.

(20) L199 TYR, or T228R ...

Thanks for your comment. Corrected.

(21) L211 and 90 ng.

Thanks for your comment. Corrected.

(22) L213 genomic DNA instead of gDNA, because gDNA may be confused in the context of sgRNA.

Thanks for your suggestion. Corrected.

(23) L253 (Fig. 1A-B).

Thanks for your comment. Corrected.

(24) L268 Explain why these 15 DEGs were selected for qRT-PCR.

Thanks for your comment. These 15 DEGs were randomly selected and act as representative DEGs with different expression levels. The reason for selection of these 15 DEGs were added in the manuscript (Lines 295-296).

(25) L287 What about GABRB? It has a TM domain.

GABRB refers to “gamma-aminobutyric acid receptor subunit beta-like” annotated on NCBI. Theoretically, it should contain four transmembrane structural domains, while it has only one, indicating that it is incomplete.

(26) L297 Add dsGFP as another control group.

Thanks for your comment. Corrected.

(27) L299 increased by 30.44% (Remove a comma).

Thanks for your comment. Corrected.

(28) L308 XM_022318019.1 (or protein accession number with XP_).

Thanks for your comment. Corrected.

(29) L338 that THR228 was conserved only in Hemiptera.

Thanks for your comment. Since our original intention was to emphasize that THR228 is the only conserved among the four key amino acid residues, after careful consideration, we retained the expression "only THR228".

(30) L342 or T228R.

Thanks for your comment. Corrected.

(31) L382 Is pyrhidone a general name for pymetrozine?

Thanks for your comment. Corrected.

(32) L450 Remove "and so on".

Thanks for your comment. Corrected.

(33) Figure 1D: Remove "Environment friendly". Replace the plant pot image on the right side with the one sprayed with pymetrozine, like the one in Figure 1F.

Thanks for your comment.

(a) "Environment friendly" in Figure 1D has been removed.

(b) We have attempted to modify the Figure 1D according to your suggestion. However, the modified Figure 1D is similar to Figure 1F and appears monotonous. Therefore, we have retained the original framework of Figure 1D.

(34) Figure 2E 111036117 and 111041856 are in different IDs (XM_). I suggest keeping GeneID in Figure 2E and Table S2, as shown in Table S4.

Thanks for your comment. Corrected.

(35) Figure 2H: Add unit of the heatmap values. Or just add the title (e.g., expression level) on top of the bar.

Thanks for your comment. Corrected.

(36) Figure 3A: Add "aa" next to 700.

Thanks for your comment. Corrected.

(37) Figure 3E-G: Revise the tick marks on Y-axis: 0.0, 0.5, 1.0, and 1.5.

Thanks for your comment. Corrected.

(38) Figure 5C: Remove "1" and move "WT" up to the position where "1" was.

Thanks for your comment. Corrected.

(39) Figure 5D: Revise the tick marks on the Y-axis: 0.0, 0.5, 1.0, and 1.5.

Thanks for your comment. Corrected.

(40) Figure 5E: Remove the decimal. (e.g. 5 uM, 10 uM, 20 uM, etc.).

Thanks for your comment. Corrected.

(41) Figure 6B: What are the numbers next to the amino acid sequences? Provide the information in the figure caption.

Thanks for your comment. The numbers next to the amino acid indicates the site of the last residue of the key amino acids, which was supplied in the figure caption.

(42) Figure 6D: Revise the tick marks on the Y-axis: 0.0, 0.5, 1.0, and 1.5. The X-axis title should be betulin (see Figure 5D). In the figure caption at the 5th row from the top, R244A should be R224A.

Thanks for your comment. Corrected.

(43) Figure 7E: R122T (not R1272T).

Thanks for your comment. Corrected.

(44) Supplementary Figure 1: It should be Figure S1. Add dsGFP in the figure caption.

Thanks for your comment. Corrected.

(45) Figure S2: What are the two pink bars and the other bars in brown or blue? Add an appropriate explanation in the figure caption.

Thanks for your comment. Corrected.

(46) Table S1: r square?

Thanks for your comment. It is “r square” and corrected.

(47) Table S2: (a) Add horizontal lines to separate qPCR, RNAi, cloning, and heterologous expression from each other (b) Replace XM_022318017.1 and XM_022318019.1 with their corresponding GeneIDs, as shown in Table S4. (c) AK340444.1 is a sequence from another aphid (Acyrthosiphon pisum)-Revise it. (d) In the cloning primers, place MpGABR first, followed by MpGABRAP and MpGABRB, as shown in the manuscript and Table S5. (e) Also, in the cloning primers, MpGABRB and MpGABRAP use reverse primers without stop codon, while MpGABR uses stop codon (TCA = TGA in reverse)-Revise it accordingly. Otherwise, provide the reason.

Thanks for your comment. Corrected.

(48) Table S3: (a) Add "*Drosophila melanogaster*" and the target sequence ID in the table caption. Is it KF881792.1, as shown in Table S6? (b) Align the sequences to the left side.

Thanks for your comment.

(a) The GenBank number of target sequence is KF881792.1 (*Drosophila melanogaster*). We have added this information in the Table S3 note.

(b) It has been adjusted according to your suggestion.

(49) Table S5: (a) Replace the accession numbers with GeneID, as shown in Table S4. K340444.1 is a sequence from another aphid (Acyrthosiphon pisum), (b) Coding sequences with stop codon are 2082, 357, and 753, respectively, while the sequences without stop codon are 2079, 354, and 750, respectively. The lengths of the deduced amino acids are 693, 118, and 250. Revise accordingly.

Thanks for your comment. Corrected.

(50) Table S6: (a) Use GenBank No for protein sequences. There is no Gene ID in this table. (b) Order (instead of Class). (c) See my comment on the phylogenetic analysis above.

Thanks for your comment. Corrected.

(51) Table S7 (a) Add unit under "Binding Energy". (b) There are two ALA226 [Alkyl] with two different distances. (c) PHE227 at the bottom should be THR228?

Thanks for your comment.

(a) The unit of "Binding Energy" was kcal.mol^–1^, and it was added in the table caption.

(b) Refer to Figure 6A, there were two Alkyl interaction between ALA226 and betulin. Therefore, there were two ALA226 [Alkyl] with two different distances.

(c) Similarly, there were two Pi-Alkyl interactions between PHE227 and betulin. Thus, there were two rows of PHE227 in the table.

(52) Table S9 (a) R117T should be R122T. (b) r square?

Thanks for your comment. a and b Corrected.

**Reviewer #2 (Recommendations for the authors):**
(1) Introduction(a) It lacks a deeper biological and evolutionary framing of the GABA receptor system. As GABA receptors are highly conserved across animal taxa, the observed interaction between betulin and the aphid GABA receptor could have broader implications. This possibility is not addressed in the current version, which limits the reader's appreciation of the relevance of this mode of action.(b) Previous reports of betulin activity in mammalian systems are not mentioned in the introduction, even though they are directly relevant to concerns about off-target toxicity. Therefore, the introduction should be revised to (i) briefly introduce the evolutionary conservation of GABA receptors, and (ii) acknowledge that betulin may affect a broader range of organisms, which sets up the need for caution in its application.

Thanks for your important suggestions.

(a) Briefly introduce the evolutionary conservation of GABA receptors has been added in the Introduction (Lines 90-98): Previous study has proposed that vertebrate and human GABR genes maintain a broad and conservative gene clustering pattern, while in invertebrates, this pattern is missing, indicating that these gene clusters formed early in vertebrate evolution and were established after diverging from invertebrates. Notably, invertebrates each possess a unique GABR gene pair, which are homologous with human GABR α and β subunits, suggesting that the existing GABR gene cluster evolved from an ancestral α - β subunit gene pair (Tsang et al. 2006). During the coevolution of plants and insects, the duplications and amino acid substitutions in GABR may be beneficial for the adaptation to insecticides and terpenoid compounds (Guo et al. 2023).

(b) The possible effects of betulin on a broader range of organisms have been acknowledged in the Introduction section (Lines 68-77): An immune stimulant, Ir-Bet, was prepared using iridium complex and betulin, which evoked ferritinophagy-enhanced ferroptosis, thereby activating anti-tumor immunity (Lv 2023). The anti-inflammatory effect of betulin has been reported in macrophages at lymphoma site in mice (Szlasa et al. 2023). Betulin has been found to improve hyperlipidemia and insulin resistance and decrease atherosclerotic plaques by inhibiting the maturation of sterol regulatory element-binding protein (Tang et al. 2011). Besides, betulin and its derivatives have been found to exhibit insecticidal activity against Plutella xylostella L. (Huang et al. 2025), Aedes aegypti (de Almeida Teles et al. 2024), and *Drosophila melanogaster* (Lee and Min 2024).

(c) At the end of the introduction, we remind that betulin should be used with caution (Lines 111-112): However, given that betulin may affect a wider range of organisms, it should be used with caution.

(2) MethodNumber of biological replicates in all assays and justification of thresholds used for significance in RNAi and survival experiments are not addressed in the manuscript.

Thanks for your careful reading. We have checked Materials and Methods section and added corresponding number of biological replicates in all assays. Besides, the p-values for the corresponding significance analyses of RNAi and survival experiments have been added to our Manuscript.

(2) Discussion(a) Consistent with the comments on the Introduction, the absence of discussion on (i) the evolutionary conservation of GABA receptor sensitivity to betulin, (ii) potential off-target effects in non-target insects and vertebrates (if so, this cannot be use for "eco-friendly pesticide" as the authors stated in the manuscript), and (iii) ecological risks associated with the exogenous application of betulin limits both the interpretive depth and applied relevance of the study.(b) To strengthen the Discussion, the authors should consider addressing: (i) whether the observed sensitivity reflects a conserved pharmacological vulnerability across animal taxa or a lineage-specific adaptation; (ii) the potential ecological risks of deploying betulin as a bioinsecticide, and (iii) the need for future research into the environmental fate, degradation, and safety profile of betulin prior to any field-level application.

Thank you for your valuable comments.

(a) We have added the discussion of the sensitivity of GABA receptor to betulin in Discussion section (Lines 491-501): Studies on key amino acids that are crucial for GABR function has primarily focused on transmembrane regions. For instance, based on the mutational research and Drosophila GABR modeling approach, multiple key amino acids were identified as insecticide targets in the transmembrane domain (Nakao and Banba 2021). Guo et al. proposed that amino acid substitutions in the transmembrane domain 2 contribute to terpenoid insensitivity during plant-insect coevolution (Guo et al. 2023). However, these studies have neglected the extracellular domain. Our study signified that betulin targets the THR228 site in the extracellular domain of MpGABR, which is conserved only in the Aphididae family. Therefore, betulin is speculated to be a specific insecticidal substance evolved by plants in response to aphid infestation. Besides, further verification is needed to determine whether betulin is toxic to other insect species.

(b) The discussion of potential ecological risks of deploying betulin as a bioinsecticide has been elaborated in our manuscript (Lines 538-551): The development of bioinsecticides should not only focus on the toxic effects of active substance on target organisms, but also on their influence on the ecosystem (Haddi et al. 2020). Although our results indicate that betulin had specific toxicity to aphids, previous studies have reported that betulin and its derivatives had effects on Plutella xylostella L. (Huang et al. 2025), Aedes aegypti (de Almeida Teles et al. 2024), and *Drosophila melanogaster* (Lee and Min 2024). Therefore, further research is needed to determine whether there are other insecticidal mechanisms or off target effects of betulin. Additionally, betulin exhibits a wide range of pharmacological activities (Amiri et al. 2020), which have been used to treat various diseases, such as cancer (Lv 2023), glioblastoma (Li et al. 2022), inflammation (Szlasa et al. 2023) and hyperlipidemia (Tang et al. 2011). Before applying betulin in the field, it is necessary to fully verify and consider whether betulin has any impact on farmers' health. Furthermore, will betulin cause residue or diffusion in the process of field application? Will long-term application promote the evolution of resistance to aphids or other insects? These issues also need further experimental verification.

(c) Additionally, at the end of the Discussion, we remind that more research is needed before any field application of betulin (Lines 551-553): In summary, before any field application, further research on the environmental behavior, degradation process, and safety of betulin is needed.

Reference

Amiri S, Dastghaib S, Ahmadi M, Mehrbod P, Khadem F, Behrouj H, Aghanoori M, Machaj F, Ghamsari M, Rosik J, Hudecki A, Afkhami A, Hashemi M, Los M, Mokarram P, Madrakian T, Ghavami S. 2020. Betulin and its derivatives as novel compounds with different pharmacological effects. Biotechnology Advances 38: 107409.

de Almeida Teles AC, dos Santos BO, Santana EC, Durço AO, Conceição LSR, Roman Campos D, de Holanda Cavalcanti SC, de Souza Araujo AA, dos Santos MRV. 2024.

Larvicidal activity of terpenes and their derivatives against Aedes aegypti: a systematic review and meta-analysis. Environmental Science and Pollution Research 31: 64703-64718.

Guo L, Qiao X, Haji D, Zhou T, Liu Z, Whiteman NK, Huang J. 2023. Convergent resistance to GABA receptor neurotoxins through plant–insect coevolution. Nature Ecology & Evolution 7: 1444-1456.

Haddi K, Turchen LM, Viteri Jumbo LO, Guedes RN, Pereira EJ, Aguiar RW, Oliveira EE. 2020. Rethinking biorational insecticides for pest management: unintended effects and consequences. Pest Management Science 76: 2286-2293.

Huang X, Hao N, Shu L, Wei Z, Shi J, Tian Y, Chen G, Yang X, Che Z. 2025. Preparation and insecticidal activities of betulin-cinnamic acid-related hybrid compounds and insights into the stress response of Plutella xylostella L. Pest Management Science 81: 4243-4255.

Lee HY, Min KJ. 2024. Betulinic acid increases the lifespan of *Drosophila melanogaster* via Sir2 and FoxO activation. Nutrients 16: 441.

Li Q, Wang L, Tang C, Wang X, Yu Z, Ping X, Ding M, Zheng L. 2024. Adipose tissue exosome circ_sxc mediates the modulatory of adiposomes on brain aging by inhibiting brain dme-miR-87-3p. Molecular Neurobiology 61: 224-238.

Li Y, Wang Y, Gao L, Tan Y, Cai J, Ye Z, Chen A, Xu Y, Zhao L, Tong S, Sun Q, Liu B, Zhang S, Tian D, Deng G, Zhou J, Chen Q. 2022. Betulinic acid self-assembled nanoparticles for effective treatment of glioblastoma. Journal of Nanobiotechnology 20: 39.

Liu S, Lamaze A, Liu Q, Tabuchi M, Yang Y, Fowler M, Bharadwaj R, Zhang J, Bedont J,

Blackshaw S, Lloyd Thomas E, Montell C, Sehgal A, Koh K, Wu Mark N. 2014. WIDE AWAKE mediates the circadian timing of sleep onset. Neuron 82: 151-166.

Lund IV, Hu Y, Raol YH, Benham RS, Faris R, Russek SJ, Brooks Kayal AR. 2008. BDNF selectively regulates GABAA receptor transcription by activation of the JAK/STAT pathway. Science Signaling 1: ra9.

Lv M, Zheng Y, Wu J, Shen Z, Guo B, Hu G, Huang Y, Zhao J, Qian Y, Su Z, Wu C, Xue X, Liu H, Mao Z. 2023. Evoking ferroptosis by synergistic enhancement of a cyclopentadienyl iridium-betulin immune agonist. Angewandte Chemie International Edition 62: e202312897.

Nakao T, Banba S. 2021. Important amino acids for function of the insect Rdl GABA receptor. Pest Management Science 77: 3753-3762.

Pope SD, Medzhitov R. 2018. Emerging principles of gene expression programs and their regulation. Molecular Cell 71: 389-397.

Szlasa W, Ślusarczyk S, Nawrot Hadzik I, Abel R, Zalesińska A, Szewczyk A, Sauer N, Preissner R, Saczko J, Drąg M, Poręba M, Daczewska M, Kulbacka J, Drąg Zalesińska M. 2023. Betulin and its derivatives reduce inflammation and COX-2 cctivity in macrophages. Inflammation 46: 573-583.

Tang JJ, Li JG, Qi W, Qiu WW, Li PS, Li BL, Song BL. 2011. Inhibition of SREBP by a small molecule, betulin, improves hyperlipidemia and insulin resistance and reduces atherosclerotic plaques. Cell Metabolism 13: 44-56.

Tsang SY, Ng SK, Xu Z, Xue H. 2006. The evolution of GABAA receptor–like genes. Molecular Biology and Evolution 24: 599-610.